# SELF-SUPERVISED LEARNING WITH ROTATION-INVARIANT KERNELS

**Léon Zheng**[1,2]  **Gilles Puy**[1]  **Elisa Riccietti**[2]  **Patrick Pérez**[1]  **Rémi Gribonval**[2]

[1]valeo.ai, Paris, France
[2]Univ Lyon, EnsL, UCBL, CNRS, Inria, LIP, F-69342, LYON Cedex 07, France.

## ABSTRACT

We introduce a regularization loss based on kernel mean embeddings with rotation-invariant kernels on the hypersphere (also known as dot-product kernels) for self-supervised learning of image representations. Besides being fully competitive with the state of the art, our method significantly reduces time and memory complexity for self-supervised training, making it implementable for very large embedding dimensions on existing devices and more easily adjustable than previous methods to settings with limited resources. Our work follows the major paradigm where the model learns to be invariant to some predefined image transformations (cropping, blurring, color jittering, etc.), while avoiding a degenerate solution by regularizing the embedding distribution. Our particular contribution is to propose a loss family promoting the embedding distribution to be close to the uniform distribution on the hypersphere, with respect to the maximum mean discrepancy pseudometric. We demonstrate that this family encompasses several regularizers of former methods, including uniformity-based and information-maximization methods, which are variants of our flexible regularization loss with different kernels. Beyond its practical consequences for state-of-the-art self-supervised learning with limited resources, the proposed generic regularization approach opens perspectives to leverage more widely the literature on kernel methods in order to improve self-supervised learning methods.

## 1 INTRODUCTION

Self-supervised learning is a promising approach for learning visual representations: recent methods (He et al., 2020; Grill et al., 2020; Caron et al., 2020; Gidaris et al., 2021) reach the performance of supervised pretraining in terms of quality for transfer learning in many downstream tasks, like classification, object detection, semantic segmentation, etc. These methods rely on some prior knowledge on images: the semantic of an image is invariant (Misra & Maaten, 2020) to some *small* transformations of the image, such as cropping, blurring, color jittering, etc. One way to design an objective function that encodes such an invariance property is to enforce two different augmentations of the same image to have a similar representation (or *embedding*) when they are encoded by the neural network. However, the main issue with this kind of objective function is to avoid an undesirable loss of information (Jing et al., 2022) where, e.g., the network learns to represent all images by the same constant representation. Hence, one of the main challenges in self-supervised learning is to propose an efficient way to *regularize* the embedding distribution in order to avoid such a *collapse*.

Our contribution is to propose a generic regularization loss promoting the embedding distribution to be close to the uniform distribution on the hypersphere, with respect to the maximum mean discrepancy (MMD), a distance on the space of probability measures based on the notion of embedding probabilities in a reproducing kernel Hilbert space (RKHS), using the so-called *kernel mean embedding* mapping. Inspired by high-dimensional statistical tests for uniformity that are rotation-invariant (García-Portugués & Verdebout, 2018), we choose to embed probability distributions using *rotation-invariant* kernels on the hypersphere (dot-product kernels), i.e., kernels for which the evaluation for two vectors depends only on their inner product (Smola et al., 2000). This paper shows that such an approach leads to important theoretical and practical consequences for self-supervised learning.

Code: https://github.com/valeoai/sfrik

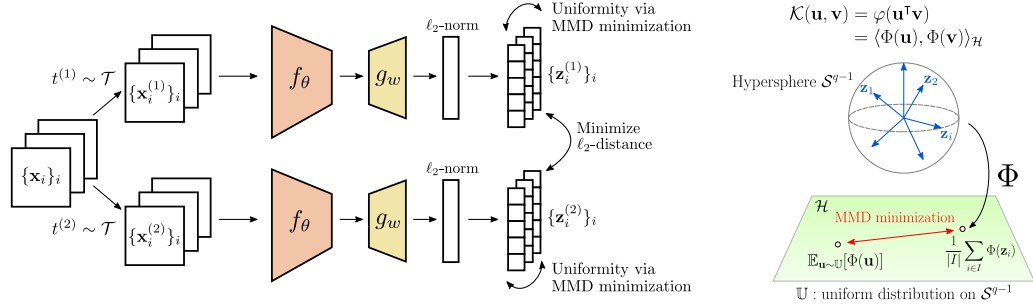

Figure 1: **Self-supervised learning with rotation-invariant kernels**. The invariance criterion minimizes the $\ell_2$-distance between two normalized embeddings $\{\mathbf{z}_i^{(v)}\}_{v=1,2}$ of two views of the same image $\mathbf{x}_i$ encoded by the backbone $f_\theta$ and the projection head $g_w$. To avoid collapse, the embedding distribution is regularized to be close to the uniform distribution on the hypersphere, in the sense of the MMD associated to a rotation-invariant kernel $\mathcal{K}(\mathbf{u}, \mathbf{v}) = \varphi(\mathbf{u}^\top \mathbf{v})$ defined on the hypersphere.

We demonstrate that our regularization loss family parameterized by such rotation-invariant kernels encompasses several regularizers of former methods. As illustrated in Table 1, they are variants of our generic loss with different kernels: the quadratic kernel yields the general sample-contrastive criterion of Garrido et al. (2023) that englobes many contrastive learning methods like (HaoChen et al., 2021) (cf. Appendix A.2); the radial basis function (RBF) kernel yields the uniformity loss of Alignment & Uniformity on the Hypersphere (AUH) (Wang & Isola, 2020); the generalized distance kernel (cf. Example 2) yields one of the regularization used in PointContrast (Xie et al.,

Table 1: Correspondence between kernel choices $\mathcal{K}(\cdot, \cdot)$ in our generic regularization loss and regularizers of former methods.

| $\mathcal{K}(\mathbf{u}, \mathbf{v})$ | Method |
|---|---|
| $(\mathbf{u}\mathbf{v}^\top)^2$ | Contrastive |
| $e^{-t\|\mathbf{u}-\mathbf{v}\|_2^2}$ | AUH |
| $C - \|\mathbf{u} - \mathbf{v}\|_2^{2s-q+1}$ | PointContrast |
| $b_1 \mathbf{u}\mathbf{v}^\top + b_2 \frac{q(\mathbf{u}\mathbf{v}^\top)^2 - 1}{q-1}$ | Analog to VICReg (cf. Section 3.3) |

2020); and a linear combination of the linear kernel and the quadratic kernel yields a regularizer that promotes the covariance matrix of the embedding distribution to be proportional to the identity matrix, similarly to information-maximization methods like VICReg (Bardes et al., 2022). In other words, these former methods turn out to be particular ways of minimizing the MMD between the embedding distribution and the uniform distribution on the hypersphere during training, with various specific kernel choices. The proposed generic regularization approach opens perspectives to leverage more widely the literature on kernel methods in order to improve self-supervised learning.

Numerically, we show in a rigorous experimental setting with a separate validation set for hyperparameter tuning that our method yields fully competitive results compared to the state of the art, when choosing truncated kernels of the form $\mathcal{K}(\mathbf{u}, \mathbf{v}) = \sum_{\ell=0}^{L} b_\ell P_\ell(q; \mathbf{u}^\top \mathbf{v})$, with $L \in \{2, 3\}$, $b_\ell \geq 0$ for $\ell \in \{0, \dots, L\}$, where $P_\ell(q; \cdot)$ denotes the Legendre polynomial of order $\ell$, dimension $q$. To our knowledge, this kernel choice has not been considered in previous self-supervision methods. Therefore, we introduce SFRIK (SelF-supervised learning with Rotation-Invariant Kernels, pronounced like "spheric"), which regularizes the embedding distribution to be close to the uniform distribution with respect to the MMD associated to such a truncated kernel, as summarized in Figure 1.

Importantly, our method significantly reduces time and memory complexity for self-supervised training compared to information-maximization methods. Due to the kernel trick, the complexity of SFRIK's loss is quadratic in the batch size and *linear* in the embedding dimension, instead of being quadratic as in VICReg. In practice, SFRIK's pretraining time is up to 19% faster than VICReg for an embedding dimension 16384, and it can scale at dimension 32768, as opposed to VICReg whose memory requirement is too large at this dimension for a machine with 8 GPUs and 32GB of memory per GPU. Hence our work opens perspectives in self-supervised learning on embedded devices with limited memory like in (Xiao et al., 2022). We summarize our contributions as follows:

- We introduce a generic regularization loss based on kernel mean embeddings with rotation-invariant kernels on the hypersphere for self-supervised learning of image representations.
- We show that our loss family encompasses several previous self-supervised learning methods, like uniformity-based and information-maximization methods.
- We numerically show that SFRIK significantly reduces time and memory complexity for self-supervised training, while remaining fully competitive with the state of the art.

## 2 RELATED WORK

Instance discrimination methods typically rely on a contrastive loss that behaves asymptotically like an alignment and uniformity loss on the hypersphere in the limit of infinite samples. Our contribution is to formalize and generalize existing uniformity-based methods by using kernel mean embeddings. To the best of our knowledge, the proposed kernel framework establishes the first connection between uniformity-based methods and information-maximization methods like VICReg.

**Instance discrimination**    One way of learning image representations that are *invariant* to predefined image transformations (Misra & Maaten, 2020) is to rely on an instance classification approach (Wu et al., 2018). Typically, contrastive learning (Oord et al., 2018; Hjelm et al., 2019; Chen et al., 2020a;b; He et al., 2020; Henaff, 2020) discriminates instances within a batch of sampled images using the noise contrastive estimator (Gutmann & Hyvärinen, 2010), by attracting embeddings of transformed images coming from the same image instance, and repulsing embeddings coming from different image instances. In practice, this estimator needs a large number of image representations in order to achieve good results, which requires a large batch size like SimCLR (Chen et al., 2020a) or a memory bank (Wu et al., 2018; He et al., 2020). In the limit of infinite samples, the contrastive loss is shown to behave asymptotically like the alignment and uniformity loss of AUH.

**Uniformity on the hypersphere**    Existing uniformity-based methods avoid collapse by regularizing the embedding distribution to be somehow close to the uniform distribution on the hypersphere, which has a high entropy. Bojanowski & Joulin (2017) perform this kind of regularization by aligning the learned representations on a fixed number of vectors sampled uniformly at random on the hypersphere. AUH maximizes the average pairwise distance between embeddings using an RBF kernel, in the spirit of energy minimization methods that address the problem of scattering points evenly on the hypersphere (Hardin & Saff, 2005; Liu et al., 2018; Borodachov et al., 2019). Although alternative high-entropy prior distributions (e.g., the uniform distribution on the hypercube) can be used for regularization (Chen et al., 2021), encoding images into $\ell_2$-normalized representations helps to stabilize training (Schroff et al., 2015; Parkhi et al., 2015; Liu et al., 2017).

**Kernel mean embedding**    As a contribution, our generic loss formalizes and generalizes these previous uniformity losses, by relying on kernel mean embeddings (cf. Appendix A.1) to measure the distance between probability distributions on high-dimensional spaces, using the MMD pseudometric (Gretton et al., 2012; Li et al., 2015; Dziugaite et al., 2015; Briol et al., 2019) with rotation-invariant kernels on the hypersphere (Smola et al., 2000; Pennington et al., 2015; Lyu, 2017; Dutordoir et al., 2020). These tools are adapted for high-dimensional problems on the hypersphere whose geometry is different from the one in small dimension, as illustrated by García-Portugués & Verdebout (2018): many statistical tests for uniformity on the hypersphere, i.e., tests for rejecting the null hypothesis where a batch of normalized vectors is sampled from the uniform distribution on the hypersphere, are in fact precisely estimators of the MMD between the embedding distribution and the uniform distribution, for different kernels. Our kernel method for self-supervision is complementary to (Li et al., 2021), in which the dependency between image instances and their embedding is maximized with respect to the Hilbert-Schmidt independence criterion (cf. Appendix A.3).

**Information maximization**    Our generic regularization approach has the benefit of connecting uniformity-based and information-maximization methods (Zbontar et al., 2021; Ermolov et al., 2021; Bardes et al., 2022). The latter are alternatives to distillation methods (Grill et al., 2020; Gidaris et al., 2020; 2021; Chen & He, 2021; Caron et al., 2021) where a student network learns to predict the representations of a teacher network. In such methods, using various architecture tricks (like prediction head, stop-gradient, momentum encoder, batch normalization or centering) is shown empirically to be sufficient to avoid collapse without instance discrimination, even though it is not fully understood how these multiple factors induce a regularization during training (Richemond et al., 2020; Tian et al., 2021). Instead of using these tricks, information-maximization methods use a Siamese architecture and avoid collapse by maximizing the statistical information of a batch of embeddings, using a whitening operation (Ermolov et al., 2021), or an explicit regularization term making the covariance (Bardes et al., 2022) or the cross-correlation (Zbontar et al., 2021) matrix close to a scaled identity matrix. This paper shows that our generic regularization loss with an appropriate kernel also promotes the covariance matrix of the embedding distribution to be proportional to the identity matrix. But in contrast to VICReg which explicitly computes the covariance matrix, our method uses the kernel trick to significantly reduce complexity at large embedding dimensions.

## 3    METHOD DESCRIPTION

Given an unlabeled dataset of images $\mathbf{x}_i \sim \mathbb{P}$, $i \in [N] := \{1, \ldots, N\}$, sampled independently from a data distribution $\mathbb{P}$, the goal is to learn *a backbone* network $f_\theta$ parameterized by $\theta$ (e.g., a convolutional neural network) such that any new image $\mathbf{x} \sim \mathbb{P}$ is encoded by a good representation $f_\theta(\mathbf{x})$ whose quality is evaluated in several downstream tasks (see Section 4).

### 3.1    INVARIANCE AND UNIFORMITY FOR SELF-SUPERVISION

Our self-supervised learning method (see Figure 1) follows the principle of the recent methods like SimCLR or VICReg. During self-supervised training, each image $\mathbf{x}_i$ is augmented using two different random transformations $t^{(1)}$ and $t^{(2)}$ sampled from a distribution $\mathcal{T}$, which yields two views $\mathbf{x}_i^{(1)} := t^{(1)}(\mathbf{x}_i)$ and $\mathbf{x}_i^{(2)} := t^{(2)}(\mathbf{x}_i)$ of the image $\mathbf{x}_i$. Two representations $\mathbf{z}_i^{(v)}$ ($v = 1, 2$) are obtained by encoding each $\mathbf{x}_i^{(v)}$ with the backbone $f_\theta$ and $\ell_2$-normalizing the resulting feature vector. For a given subset of indices $I \subseteq [N]$, we write $\mathbf{Z}_I^{(v)} := \{\mathbf{z}_i^{(v)}\}_{i \in I}$. The backbone $f_\theta$ is trained by minimizing the total objective function:

$$\mathcal{L} = \mathbb{E}_{t^{(1)}, t^{(2)} \sim \mathcal{T}} \, \mathbb{E}_{I \subseteq [N]} \, \ell(\mathbf{Z}_I^{(1)}, \mathbf{Z}_I^{(2)}), \tag{1}$$

where batches $I$ are drawn at random with a prescribed batch size, and the loss $\ell$ is a weighted sum involving an alignment term $\ell_a$ and a uniformity term $\ell_u$, in the spirit of AUH:

$$\ell(\mathbf{Z}_I^{(1)}, \mathbf{Z}_I^{(2)}) := \lambda \, \ell_a(\mathbf{Z}_I^{(1)}, \mathbf{Z}_I^{(2)}) + 0.5 \, (\ell_u(\mathbf{Z}_I^{(1)}) + \ell_u(\mathbf{Z}_I^{(2)})) \, ; \tag{2}$$

$\lambda > 0$ is a hyperparameter that tunes the balance between the two terms. The loss $\ell_a$ enforces the invariance property of the model, and is defined for a batch $I \subseteq [N]$ of cardinality $|I|$ as:

$$\ell_a(\mathbf{Z}_I^{(1)}, \mathbf{Z}_I^{(2)}) := \frac{1}{|I|} \sum_{i \in I} \left\| \mathbf{z}_i^{(1)} - \mathbf{z}_i^{(2)} \right\|_2^2. \tag{3}$$

Our main contribution is in the choice of the uniformity term $\ell_u$, detailed in the rest of the section. Note that instead of applying the loss (2) to the output of $f_\theta$ (called image representation), we add a projection head $g_w$ (a multi-layer perceptron) parameterized by $w$ to the output of $f_\theta$ and apply (2) at the output of $g_w$ (called image embedding). This common practice (Caron et al., 2020; Grill et al., 2020) improves the performance in the downstream tasks. Therefore, denoting $\mathcal{S}^{q-1}$ the unit hypersphere in $\mathbb{R}^q$, the image embedding actually reads $\mathbf{z}_i^{(v)} := (g_w \circ f_\theta)(\mathbf{x}_i^{(v)}) / \left\| (g_w \circ f_\theta)(\mathbf{x}_i^{(v)}) \right\|_2 \in \mathcal{S}^{q-1}$. Both $g_w$ and $f_\theta$ are jointly trained without supervision by minimizing (1) using a stochastic mini-batch algorithm. After training, $g_w$ is discarded and only $f_\theta$ is kept for the downstream tasks.

### 3.2    UNIFORMITY LOSS VIA MMD MINIMIZATION

We continue by explaining our generic kernel formulation of $\ell_u$ using the MMD pseudometric and rotation-invariant kernels. Then we provide examples of such kernels and describe our kernel choice.

#### 3.2.1    MMD PSEUDOMETRIC AND ROTATION-INVARIANT KERNELS

Our uniformity loss relies on a divergence in the space of probability distributions based on a positive definite kernel $\mathcal{K}$ defined on some space $\mathcal{X}$. Denoting $\mathcal{H}$ the corresponding RKHS with norm $\|\cdot\|_\mathcal{H}$, the MMD between two probability distributions $\mathbb{Q}_1, \mathbb{Q}_2$ on $\mathcal{X}$ can be expressed as the distance in $\|\cdot\|_\mathcal{H}$ between their kernel mean embeddings (Borgwardt et al., 2006; Muandet et al., 2017):

$$\mathrm{MMD}(\mathbb{Q}_1, \mathbb{Q}_2) = \left\| \int_\mathcal{X} \mathcal{K}(\mathbf{u}, \cdot) d\mathbb{Q}_1(\mathbf{u}) - \int_\mathcal{X} \mathcal{K}(\mathbf{u}, \cdot) d\mathbb{Q}_2(\mathbf{u}) \right\|_\mathcal{H}. \tag{4}$$

We propose to use this pseudometric to measure the distance between the probability distribution of the embeddings $\mathbf{z}_i^{(v)}$ ($v = 1, 2$) and the uniform probability distribution on the hypersphere $\mathcal{S}^{q-1}$ defined by $\mathbb{U} := \sigma_{q-1} / |\mathcal{S}^{q-1}|$, where $\sigma_{q-1}$ denotes the normalized Hausdorff surface measure on $\mathcal{S}^{q-1}$, and $|\mathcal{S}^{q-1}| := \int_{\mathcal{S}^{q-1}} d\sigma_{q-1} = 2\pi^{\frac{q}{2}} / \Gamma(\frac{q}{2})$ is the surface area of $\mathcal{S}^{q-1}$, with $\Gamma$ denoting the Gamma function. Intuitively, a good choice of kernel for measuring the distance (4) should

distinguish any probability distribution from the uniform distribution. Inspired by statistical tests for uniformity that are rotation-invariant (García-Portugués & Verdebout, 2018), we propose to use *rotation-invariant* kernels on $\mathcal{X} := \mathcal{S}^{q-1}$ of the form $\mathcal{K}(\mathbf{u}, \mathbf{v}) := \varphi(\mathbf{u}^\top \mathbf{v})$ with $\varphi$ a continuous function defined on $[-1, 1]$ (Smola et al., 2000). The following theorem characterizes the form of function $\varphi$ that ensures positive definiteness of $\mathcal{K}$, and thus that (4) is a valid pseudometric.

**Theorem 1 (Schoenberg (1942, Theorem 1))** *The kernel $\mathcal{K}(\mathbf{u}, \mathbf{v}) := \varphi(\mathbf{u}^\top \mathbf{v})$ on $\mathcal{X} := \mathcal{S}^{q-1}$ with $\varphi$ continuous is positive definite if, and only if, the function $\varphi$ admits an expansion:*

$$\varphi(t) = \sum_{\ell=0}^{+\infty} b_\ell P_\ell(q; t), \quad with \quad b_\ell \geq 0, \tag{5}$$

*where $P_\ell(q; t) := \ell! \, \Gamma\left(\frac{q-1}{2}\right) \sum_{k=0}^{\lfloor \frac{\ell}{2} \rfloor} \left(-\frac{1}{4}\right)^k \frac{(1-t^2)^k t^{\ell-2k}}{k! \, (\ell-2k)! \, \Gamma(k+\frac{q-1}{2})}$ is the Legendre (or Gegenbauer) polynomial of degree $\ell$ in dimension $q$ (Müller, 2012, (2.32)).*

As we are interested in measuring the distance between the embedding distribution and the uniform distribution on the hypersphere $\mathbb{U}$, we compute the kernel mean embedding of $\mathbb{U}$ for a kernel satisfying the condition of Theorem 1 using the following known result used, e.g., implicitly in (Brauchart et al., 2014). As we could not locate a formal proof, we provide one in Appendix B.1.

**Lemma 2** *Let $\mathcal{K}(\mathbf{u}, \mathbf{v}) := \varphi(\mathbf{u}^\top \mathbf{v})$ be a rotation-invariant kernel on $\mathcal{X} := \mathcal{S}^{q-1}$ where $\varphi$ admits the expansion (5). The kernel mean embedding of the uniform distribution $\mathbb{U}$ on $\mathcal{S}^{q-1}$ is constant: $\int_{\mathcal{S}^{q-1}} \mathcal{K}(\mathbf{u}, \mathbf{v}) \, \mathrm{d}\mathbb{U}(\mathbf{u}) = b_0 \in \mathbb{R}$ for all $\mathbf{v} \in \mathcal{S}^{q-1}$. The kernel mean embedding of any probability distribution $\mathbb{Q}$ defined on the hypersphere satisfies: $\int_{\mathcal{S}^{q-1}} \mathcal{K}(\mathbf{u}, \cdot) \, \mathrm{d}\mathbb{Q}(\mathbf{u}) = b_0 + \int_{\mathcal{S}^{q-1}} \tilde{\mathcal{K}}(\mathbf{u}, \cdot) \, \mathrm{d}\mathbb{Q}(\mathbf{u})$, where $\tilde{\mathcal{K}}(\mathbf{u}, \mathbf{v}) := \tilde{\varphi}(\mathbf{u}^\top \mathbf{v})$ for any $\mathbf{u}, \mathbf{v} \in \mathcal{S}^{q-1}$ with $\tilde{\varphi} := \sum_{\ell=1}^{+\infty} b_\ell P_\ell(q; \cdot)$.*

Using Lemma 2 in (4) yields $\mathrm{MMD}(\mathbb{Q}, \mathbb{U}) = \| \int_{\mathcal{S}^{q-1}} \tilde{\mathcal{K}}(\mathbf{u}, \cdot) \mathrm{d}\mathbb{Q}(\mathbf{u}) \|_{\mathcal{H}}$ for any probability distribution $\mathbb{Q}$ on $\mathcal{S}^{q-1}$. Then, by the reproducing property in the RKHS $\mathcal{H}$, the squared MMD satisfies, for *any* rotation-invariant kernel $\mathcal{K}$ verifying the condition of Theorem 1:

$$\mathrm{MMD}^2(\mathbb{Q}, \mathbb{U}) = \mathbb{E}_{\mathbf{z}, \mathbf{z}' \sim \mathbb{Q}} \left[ \tilde{\mathcal{K}}(\mathbf{z}, \mathbf{z}') \right], \quad \text{with } \mathbf{z}, \mathbf{z}' \text{ i.i.d.} \tag{6}$$

### 3.2.2 Estimator of the squared MMD and kernel choices

The proposed uniformity loss $\ell_u$ for self-supervision is a biased estimator (Gretton et al., 2012) of $\mathrm{MMD}^2(\mathbb{Q}, \mathbb{U})$ in (6). Given a batch $\mathbf{Z}_I := \{\mathbf{z}_i\}_{i \in I}$ sampled from $\mathbb{Q}$, our uniformity loss is:

$$\ell_u(\mathbf{Z}_I) = \widehat{\mathrm{MMD}}^2(\mathbb{Q}, \mathbb{U}; \{\mathbf{Z}_I\}) := \frac{1}{|I|^2} \sum_{i \in I} \sum_{i' \in I} \tilde{\mathcal{K}}(\mathbf{z}_i, \mathbf{z}_{i'}) = \frac{1}{|I|^2} \sum_{i \in I} \sum_{i' \in I} \tilde{\varphi}(\mathbf{z}_i^\top \mathbf{z}_{i'}). \tag{7}$$

In our framework, any rotation-invariant kernel satisfying the condition of Theorem 1 can be used to compute (7) and train a self-supervised model by minimizing (1). The uniformity term (7) can be interpreted as an energy functional (Brauchart et al., 2014): minimizing the average pairwise energy quantified by $\tilde{\mathcal{K}}$ tends to scatter evenly the embeddings on the hypersphere. We now give examples of kernels that can be used for this uniformity term. This illustrates that our framework offers a unification of several strategies for self-supervision.

**Example 1: RBF kernel.** Using $\mathcal{K}(\mathbf{u}, \mathbf{v}) = e^{-t\|\mathbf{u}-\mathbf{v}\|_2^2}$ (with $t > 0$) in the uniformity term $\ell_u$ (7) yields the regularization term from AUH, with the only difference that AUH uses the logarithm of the energy functional as their uniformity loss.

**Example 2: Generalized distance kernel.** It is defined as $\mathcal{K}(\mathbf{u}, \mathbf{v}) := C - \|\mathbf{u} - \mathbf{v}\|_2^{2s-q+1}$ with $\frac{q-1}{2} < s < \frac{q+1}{2}$ and $C > 0$ sufficiently large (Brauchart et al., 2014). A variation of this kernel choice is, e.g., used in the hard-contrastive loss of PointContrast for self-supervision on point clouds.

**Example 3: Truncations of the Laplace-Fourier series.** A truncated kernel up to order $L$ (Brauchart et al., 2014) is a kernel $\mathcal{K}(\mathbf{u}, \mathbf{v}) = \sum_{\ell=0}^{L} b_\ell P_\ell(q; \mathbf{u}^\top \mathbf{v})$, with $b_\ell \geq 0$ for $\ell = 0, \ldots, L$. It admits a closed-form expression given by the definition of Legendre polynomials $P_\ell(q, \cdot)$ in Theorem 1, e.g., $P_1(q, t) = t$, $P_2(q, t) = \frac{qt^2-1}{q-1}$, $P_3(q, t) = \frac{(q+2)t^3-3t}{q-1}$. We explore numerically this kernel choice in Section 4, since it has never been considered in previous self-supervision methods.

The expansion of $\varphi$ in Legendre polynomials (5) for the RBF (Example 1) and the generalized distance kernel (Example 2) verifies $b_\ell > 0$ for each integer $\ell$ (see Appendix B.2). By (Micchelli et al., 2006, Theorem 10), this is a necessary and sufficient condition for a rotation-invariant kernel to be universal, and universality is a sufficient condition for injectivity of the corresponding kernel mean embedding mapping, i.e., the kernel is characteristic (Fukumizu et al., 2004). The benefit of this property is to guarantee that the uniform distribution $\mathbb{U}$ is the unique solution to the minimization problem: $\min \left\{ \mathrm{MMD}(\mathbb{Q}, \mathbb{U}) \mid \mathbb{Q} \text{ is a probability distribution on } \mathcal{S}^{q-1} \right\}$. In contrast, the truncated rotation-invariant kernel up to an order $L$ (Example 3) is not universal. Yet, our experiments in Section 4 show that truncated kernels up to order $L = 2, 3$ provide better results than, e.g., AUH whose uniformity loss is based on the RBF kernel.

In summary, the uniformity loss in our method, called SFRIK, corresponds to (7) with a truncated kernel up to order $L = 3$ and satisfies:

$$\ell_u(\{\mathbf{z}_i\}_{i \in I}) = \frac{1}{|I|^2} \sum_{i \in I} \sum_{i' \in I} \left( b_1 \mathbf{z}_i^\top \mathbf{z}_{i'} + b_2 \frac{q(\mathbf{z}_i^\top \mathbf{z}_{i'})^2 - 1}{q - 1} + b_3 \frac{(q+2)(\mathbf{z}_i^\top \mathbf{z}_{i'})^3 - 3\mathbf{z}_i^\top \mathbf{z}_{i'}}{q - 1} \right), \quad (8)$$

where $b_\ell \geq 0$, $\ell = 1, 2, 3$, are hyperparameters, and $q$ is the dimension of the image embedding $\mathbf{z}_i$.

### 3.3 Connection with information-maximization methods

We now show that choosing an appropriate kernel in the proposed uniformity term (7) leads to a regularizer that maximizes a statistical measure of information analog to the one used in VICReg. To the best of our knowledge, this is the first connection made between uniformity-based and information-maximization methods. The regularization loss of VICReg is a weighted sum between two terms:

$$v(\mathbf{Z}_I) := \frac{1}{q} \sum_{j=1}^{q} \max\left(0, \gamma - \sqrt{\mathrm{Var}(z_I^j) + \varepsilon}\right), \quad c(\mathbf{Z}_I) := \frac{1}{q} \sum_{1 \leq j \neq j' \leq q} [C(\mathbf{Z}_I)]_{j,j'}^2, \quad (9)$$

for a batch of image embeddings $\mathbf{Z}_I := \{\mathbf{z}_i\}_{i \in I}$, where $z^j$ denotes the $j$-th coordinate of a (random) vector $\mathbf{z}$ and $\varepsilon$ is a fixed small scalar. The *variance* term $v(\mathbf{Z}_I)$ enforces the empirical variance $\mathrm{Var}(z_I^j) := \frac{1}{|I|-1} \sum_{i \in I} (z_i^j - \bar{z}^j)^2$ in each coordinate $j = 1, \ldots, q$ to be above a certain threshold $\gamma^2 > 0$ (here $\bar{\mathbf{z}}$ is the empirical mean of $\mathbf{Z}_I$). The *covariance* term $c(\mathbf{Z}_I)$ enforces the non-diagonal entries of the empirical covariance matrix $C(\mathbf{Z}_I) := \frac{1}{|I|-1} \sum_{i \in I} (\mathbf{z}_i - \bar{\mathbf{z}})(\mathbf{z}_i - \bar{\mathbf{z}})^\top$ to be zero.

In order to connect VICReg and SFRIK, let us consider for simplicity a truncated kernel $\tilde{\mathcal{K}}(\mathbf{u}, \mathbf{v}) = \sum_{\ell=1}^{L} b_\ell P_\ell(q; \mathbf{u}^\top \mathbf{v})$ of order $L = 2$ (the reasoning would be the same if the kernel was not truncated), and assume $b_1, b_2 > 0$. By the addition theorem (Müller, 2012, Theorem 2, §1), there exists a feature map $\Phi : \mathcal{S}^{q-1} \to \mathbb{R}^m$ involving an orthonormal basis of spherical harmonics (homogeneous harmonic polynomials restricted to the hypersphere) of order 1 and 2 such that $\Phi(\mathbf{u})^\top \Phi(\mathbf{v}) = \tilde{\mathcal{K}}(\mathbf{u}, \mathbf{v})$. Hence, the kernel mean embedding of a distribution in the associated RKHS contains its first and second-order moments (see Appendix B.3). Therefore, denoting $N(q, \ell)$ the dimension of the space of spherical harmonics of order $\ell$, dimension $q$, and defining $\Phi_\ell : \mathcal{S}^{q-1} \to \mathbb{R}^{N(q,\ell)}, \mathbf{z} \mapsto (Y_{\ell,k}(\mathbf{z}))_{k=1}^{N(q,\ell)}$ for $\ell \in \{1, 2\}$ with

$$\begin{aligned}
\{Y_{1,k}\}_{k=1}^{N(q,1)} &:= \left\{\mathbf{u} \mapsto u^j \mid 1 \leq j \leq q\right\}, \\
\{Y_{2,k}\}_{k=1}^{N(q,2)} &:= \left\{\mathbf{u} \mapsto u^j u^{j'} \mid 1 \leq j < j' \leq q\right\} \cup \left\{\mathbf{u} \mapsto (u^j)^2 - \frac{1}{q} \mid 2 \leq j \leq q\right\},
\end{aligned} \quad (10)$$

it is possible to show (see Appendix B.3) that the squared MMD (6) can be written as

$$\mathrm{MMD}^2(\mathbb{Q}, \mathbb{U}) = a_1 \left\| \mathbf{M}_1 \mathbb{E}_{\mathbf{z} \sim \mathbb{Q}}[\Phi_1(\mathbf{z})] \right\|_2^2 + a_2 \left\| \mathbf{M}_2 \mathbb{E}_{\mathbf{z} \sim \mathbb{Q}}[\Phi_2(\mathbf{z})] \right\|_2^2, \quad (11)$$

where $a_\ell := b_\ell |\mathcal{S}^{q-1}| / N(q, \ell)$ for $\ell \in \{1, 2\}$, and $\mathbf{M}_1, \mathbf{M}_2$ are two lower triangular matrices with nonzero diagonal entries. Hence, when $\mathbb{Q}$ plays the role of the embedding distribution during self-supervised training, minimizing $\mathrm{MMD}^2(\mathbb{Q}, \mathbb{U})$ in (11) as we propose for regularizing the embedding distribution promotes its expectation $\mathbb{E}_{\mathbf{z} \sim \mathbb{Q}}[\mathbf{z}]$ and its autocorrelation matrix $\mathbb{E}_{\mathbf{z} \sim \mathbb{Q}}[\mathbf{z}\mathbf{z}^\top]$ to be close to 0 and $q^{-1}\mathbf{I}_q$ respectively, where $\mathbf{I}_q$ is the identity matrix. When $\mathrm{MMD}^2(\mathbb{Q}, \mathbb{U}) = 0$, the covariance matrix is equal to $\mathbb{E}[(\mathbf{z} - \mathbb{E}[\mathbf{z}])(\mathbf{z} - \mathbb{E}[\mathbf{z}])^\top] = \mathbb{E}[\mathbf{z}\mathbf{z}^\top] - \mathbb{E}[\mathbf{z}]\mathbb{E}[\mathbf{z}]^\top = \frac{1}{q}\mathbf{I}_q$ because $b_1, b_2 > 0$ and the two terms on the right-hand side of (11) are null.

In conclusion, the regularization both in VICReg and SFRIK induces the embedding distribution to have a covariance matrix with zero non-diagonal entries. The diagonal entries of the covariance matrix are encouraged to be equal to $1/q$ in SFRIK, and greater than $\gamma^2$ in VICReg (we recall that the image embeddings $\{\mathbf{z}_i\}_{i \in I}$ are not $\ell_2$-normalized in VICReg). However, one difference in terms of regularization behavior is that SFRIK encourages the expectation of the embedding distribution to be zero, as shown in the first term of (11). This is not the case for VICReg, as we can see in (9).

Finally, the memory and computational complexities for computing the uniformity term (8) in SFRIK are $\mathcal{O}(|I|^2)$ and $\mathcal{O}(q|I|^2)$, as opposed to $\mathcal{O}(q^2)$ and $\mathcal{O}(q^2|I|)$ for the variance and covariance terms (9) in VICReg. In the setting where SFRIK and VICReg work best, i.e., larger dimension $q$ and smaller batch size $|I|$, SFRIK has the lowest memory and computational complexities. This computational advantage is due to the kernel trick and it is illustrated in Section 4.

## 4 EXPERIMENTS

We first demonstrate numerically that the regularization loss (8) of SFRIK outperforms existing alternatives, in a rigorous experimental setting with a subset of ImageNet-1000 (Deng et al., 2009) for pretraining and a separate validation set for hyperparameter tuning. Then, we pretrain a ResNet-50 backbone (He et al., 2016) with SFRIK on the full ImageNet dataset and show competitive results compared to the state of the art, with significant computational benefits during pretraining.

### 4.1 EXPERIMENTAL SETTING

The backbone $f_\theta$ is either ResNet-18 or ResNet-50, depending on the experiment. Following Zbontar et al. (2021), the projection head $g_w$ is a three-layer MLP made of two hidden layers with ReLU activation and batch normalization (Ioffe & Szegedy, 2015), and a linear output layer. Unless otherwise specified, the size (number of neurons) of the two hidden layers is the same as the one, denoted $q$, of the output layer and the default value is $q = 8192$. The augmentations used for transforming images into views are the same as the ones used in VICReg. The backbone and the projection head are trained with a LARS optimizer (You et al., 2017). The weight decay is fixed at $10^{-6}$. The learning rate scheduling starts with 10 warm-up epochs (Goyal et al., 2017) with a linear increase from 0 to $initial\_lr = base\_lr * bs/256$, where $base\_lr$ is called the base learning rate (Goyal et al., 2017) and $bs$ is the batch size, followed by a cosine decay (Loshchilov & Hutter, 2017) with a final learning rate 1000 times smaller than $initial\_lr$. For pretraining, we consider a 20% subset of ImageNet-1000 (denoted by IN20%), like in (Gidaris et al., 2021), and 100% of ImageNet-1000 (denoted by IN100%). In IN20%, we keep all the 1000 classes but only 260 images per class.

### 4.2 SFRIK'S REGULARIZER OUTPERFORMS EXISTING ALTERNATIVE ON IN20%

Many existing self-supervision methods are based on the Siamese architecture and have the same form of training loss $\lambda \ell_a(\mathbf{Z}_I^{(1)}, \mathbf{Z}_I^{(2)}) + \mu \ell_r(\mathbf{Z}_I^{(1)}, \mathbf{Z}_I^{(2)})$. This is the case of SimCLR, AUH and VICReg, for which Appendix B.4 gives the expression of the regularization loss $\ell_r$. For SFRIK, following (2), we have $\mu = 0.5$ and $\ell_r(\mathbf{Z}_I^{(1)}, \mathbf{Z}_I^{(2)}) = \ell_u(\mathbf{Z}_I^{(1)}) + \ell_u(\mathbf{Z}_I^{(2)})$ with $\ell_u$ given by (8).

**Protocol.** To isolate the impact of $\ell_r$ on the quality of the learned representations, we (re)implement all these four methods in the setting of Section 4.1, to get rid of the influence of other design choices, like image augmentations or projection head architecture. We fix the batch size at 2048, and tune the base learning rate and hyperparameters specific to each method's loss. We also compare different embedding dimension $q \in \{1024, 2048, 4096, 8192\}$. In order to perform an extensive hyperparameter tuning by grid search of each method for fair comparisons, we choose a smaller backbone and a reduced dataset for pretraining, i.e., we pretrain a ResNet-18 on IN20% for 100 epochs with all methods. Pretrained backbones are then evaluated by linear probing trained on IN20% with labels.

**Number of hyperparameters.** Note that in total SFRIK with $L = 2$ has as many hyperparameters to tune as AUH or VICReg, and SFRIK with $L = 3$ has a *single* additional hyperparameter.

**Rigorous hyperparameter tuning.** In contrast to the common practice in the literature where hyperparameters are directly selected on the evaluation dataset, we choose to tune hyperparameters on

Table 2: **Linear probing on IN20% (top-1 accuracy)** at different embedding dimensions $q$. All methods were pretrained on IN20% with a ResNet-18 for 100 epochs. We tuned all hyperparameters specific to each method and the learning rate. Symbol $\dagger$ indicates models that we retrained ourselves.

| | SimCLR$^\dagger$ | AUH$^\dagger$ | VICReg$^\dagger$ | SFRIK | | |
| | | | | $L=1$ | $L=2$ | $L=3$ |
|---|---|---|---|---|---|---|
| $q = 1024$ | 45.2 | 45.3 | 40.6 | - | 45.2 | - |
| $q = 2048$ | 45.8 | 45.9 | 44.0 | - | 45.9 | - |
| $q = 4096$ | 46.0 | 46.7 | 44.9 | - | 46.9 | - |
| $q = 8192$ | 46.1 | 46.8 | 46.0 | 27.7 | 47.0 | **47.5** |

a *separate* validation set that consists of *another* 20% subset of the ImageNet train set. We select the hyperparameters that yield the highest top-1 accuracy obtained by weighted kNN-classification ($k = 20$) (Wu et al., 2018) on this validation set, and we finally report the evaluation results by linear probing on the usual ImageNet validation set, which is never seen during hyperparameter tuning.

**Results.** Table 2 shows that SFRIK at optimal truncation order $L = 3$ outperforms SimCLR, AUH and VICReg by at least $0.7$ points at $q = 8192$. The gain in top-1 accuracy by linear probing between SFRIK at $L = 1$ and $L = 2$ is important, but is smaller between $L = 2$ and $L = 3$. This suggests that $L > 3$ is likely to marginally improve performance, while requiring more hyperparameter tuning, which is why we did not explore $L > 3$. We also remark that all methods benefit from an increase in embedding dimension $q$, including SimCLR which was originally introduced with a smaller dimension. Appendix D.2 provides extra results for linear classification on Places205 (Zhou et al., 2014) and VOC2007 (Everingham et al., 2010) that further support our findings: SFRIK outperforms AUH while having the same pretraining complexity, and is fully competitive compared to VICReg with a reduced pretraining complexity.

**Ablation.** Table 3 confirms empirically that a truncated kernel is better than the RBF[1] or the generalized distance kernel for the uniformity term (7). During tuning we observed that the truncated kernel performs well when the weights $b_2, b_3$ in (8) are larger than $b_1$, e.g., $(b_1, b_2, b_3) = (1, 40, 40)$ for $q = 8192$. This contrasts with the RBF and the generalized distance kernel for which the weights $b_\ell$ decay polynomially with respect to $\ell$ (see Appendix B.2). This suggests that it is important to focus more on order 2, 3 than on order 1 in the Legendre expansion (5).

Table 3: **Impact of kernel choice in the uniformity term** (7). Linear probing on IN20% of ResNet-18 pretrained on IN20% for 100 epochs, at $q = 8192$.

| Kernel | Top-1 acc. |
|---|---|
| RBF | 41.3 |
| Generalized distance | 27.8 |
| Truncated, $L = 3$, cf. (8) | **47.5** |

### 4.3 Results for ResNet-50 pretrained on IN100%

**Protocol.** We pretrain a ResNet-50 with SFRIK on IN100% under the setting of Section 4.1, with a batch size of 2048. We study the impact of a larger embedding dimension in SFRIK by considering a projection head with two hidden layers of size 8192, and an output layer of size $q \in \{8192, 16384, 32768\}$. Truncation order is either $L = 2$ or $L = 3$. For comparison, we also pretrain a ResNet-50 with VICReg under the same setting with $q = 8192$. Similarly to the original paper (Bardes et al., 2022), the alignment, variance and covariance weights are respectively 25, 25, 1, and the base learning rate is 0.2 for VICReg. All pretrained backbones are evaluated by: linear probing on IN100%; linear classification on Places205 and VOC2007 in order to measure how the learned representations generalize to an unseen dataset; and semi-supervised learning with few labels of IN100% (backbones are fine-tuned for classification using 1% or 10% of labeled images).

**Computational complexity.** We show under this protocol that SFRIK's time and memory complexity during pretraining is significantly smaller than the one of VICReg for large dimensions. This allows us to scale SFRIK at dimension 16384 and even to 32768 for better results on downstream tasks.[2] We measure the peak memory per GPU during pretraining on IN100% with a batch size of 2048 and the pretraining wall time of both methods on a $8\times$ AMD Radeon Instinct MI50 32GB:

---

[1]The performance gap between AUH and the RBF kernel is only due to the presence of the logarithm in AUH (cf. Example 1). Future work could clarify the role of this logarithm for regularization in self-supervision.

[2]We recall that the time and memory complexity is identical for all methods on downstream tasks.

Table 4: **Linear classification on IN100%, Places205, VOC2007, and semi-supervised learning with few labels of IN100% (accuracy or mean average precision).** Methods are pretrained on IN100% with ResNet-50. We only include methods relying on a Siamese architecture with image augmentations limited to two views. The scores of methods marked with * are from Chen & He (2021). The score of VICReg[†] was obtained by retraining the model ourselves. For each downstream task, we highlight in bold the best score among all backbones pretrained on 200 epochs.

| Method | Epochs | Linear classification | | | | | Semi-supervised | | | |
| | | IN100% | | Places205 | | VOC07 | 1% labels | | 10% labels | |
| | | Top-1 | Top-5 | Top-1 | Top-5 | mAP | Top-1 | Top-5 | Top-1 | Top-5 |
|---|---|---|---|---|---|---|---|---|---|---|
| SimCLR* (Chen et al., 2020a) | 200 | 68.3 | - | - | - | - | - | - | - | - |
| SwAV* (Caron et al., 2020) (no multi-crop) | 200 | 69.1 | - | - | - | - | - | - | - | - |
| SimSiam (Chen & He, 2021) | 200 | 70.0 | - | - | - | - | - | - | - | - |
| VICReg[†] (Bardes et al., 2022) ($q = 8192$) | 200 | 70.0 | 89.3 | 54.1 | 83.4 | 84.9 | **49.4** | **75.1** | 65.9 | 87.2 |
| SFRIK ($L = 2, q = 8192$) | 200 | 70.1 | 89.3 | 53.8 | 83.0 | 85.1 | 46.6 | 73.3 | 65.7 | 87.3 |
| SFRIK ($L = 3, q = 8192$) | 200 | 70.2 | 89.6 | **54.5** | **83.9** | 84.6 | 46.9 | 73.6 | **66.0** | **87.7** |
| SFRIK ($L = 2, q = 16384$) | 200 | **70.3** | 89.6 | 54.3 | 83.4 | **85.2** | 46.0 | 73.0 | 65.3 | 87.2 |
| SFRIK ($L = 2, q = 32768$) | 200 | **70.3** | 89.6 | 54.1 | 83.0 | 85.0 | 46.1 | 73.0 | 65.4 | 87.3 |
| SFRIK ($L = 3, q = 32768$) | 200 | **70.3** | **89.7** | 54.4 | 83.2 | 85.1 | 46.6 | 73.0 | 65.8 | 87.5 |
| SFRIK ($L = 2, q = 8192$) | 400 | 70.8 | 89.9 | 54.4 | 83.5 | 85.7 | 47.8 | 74.3 | 66.4 | 88.0 |

- at $q = 8192$, SFRIK is 8% faster than VICReg and needs 3% less memory per GPU;

- at $q = 16384$, SFRIK is 19% faster than VICReg and needs 8% less memory per GPU;

- at $q = 32768$, SFRIK is still 2% faster than VICReg *run in the lower dimension 16384*. It only requires 30.9GB per GPU while VICReg at $q = 32768$ needs more than the available memory. Table 5 emphasizes this memory advantage at reduced batch sizes.

Table 5: **Peak memory per GPU** during pretraining of ResNet-50 on IN100% at $q = 32768$.

| Batch size | VICReg | SFRIK | (ratio) |
|---|---|---|---|
| 256 | 22.5GB | 10.3GB | (2.2) |
| 512 | 25.4GB | 13.1GB | (1.9) |
| 1024 | 31.1GB | 18.8GB | (1.7) |

**Results.** Table 4 compares methods that have the same Siamese architecture and use the same image augmentations described in our protocol. For completeness, this table is completed in Appendix D.3 by evaluation results of other existing methods such as BYOL (Grill et al., 2020), OBoW (Gidaris et al., 2020) and SwAV with multi-crop (Caron et al., 2020), which are not comparable to the methods of Table 4 as they use a teacher-student architecture with momentum encoder and/or image augmentations with multi-scale cropping, and are beyond the setting of a Siamese architecture with only two views. Incorporating such designs in SFRIK is possible, and is left as a future work.

Table 4 demonstrates the competitiveness of SFRIK: it has the best accuracy for linear probing on IN100% among SimCLR, SwAV with no multi-crop, SimSiam and VICReg, and it performs better than VICReg for linear classification on Places205, VOC2007, and semi-supervised-learning with 10% of labels. We observe that SFRIK and VICReg offer a different trade-off between performance on linear probing on IN100% and performance on semi-supervised learning with 1% of labels. But as shown in Appendix D.3, other methods like BYOL and SwAV with multi-crop similarly have a performance drop compared to VICReg on semi-supervised learning with 1% of labels, even though they perform better on linear probing. Future work will therefore involve understanding what specific ingredients of VICReg make it more robust for semi-supervised learning with few labels. Ideally we could combine these ingredients with our generic kernel framework to design an improved version of SFRIK that can still benefit from its computational advantages over VICReg.

# 5 CONCLUSION

We proposed a regularization loss family based on the MMD and rotation-invariant kernels. We demonstrated that several regularizers of former methods are indeed variants of our flexible loss with different kernels. This generic regularization approach allowed us to leverage degrees of freedom in rotation-invariant kernel design to improve self-supervision methods. In practice, using a truncated kernel, we derived from the proposed framework a fully competitive self-supervised pretraining method, SFRIK, which significantly reduces time and memory complexity during pretraining compared to information-maximization methods. Combining the approach with kernel approximation techniques such as quadrature rules or random feature expansions offers promising perspectives to further enhance the ability to perform self-supervised training with limited computational resources.

ETHICS STATEMENT

The authors are concerned by the carbon footprint of deep learning research. To raise awareness of this issue in the community, we report the energy used for the computational resources of this project, which is approximately 12500 kWh.

REPRODUCIBILITY STATEMENT

In the interest of reproducible research, we provide our code and our pretrained ResNet-50 backbones with SFRIK on IN100% at https://github.com/valeoai/sfrik (Zheng et al., 2022). All details about our experimental setting can be found in either Section 4 or Appendix C. For the experiments on IN20%, as detailed in Appendix C.4, hyperparameters are tuned on a *separate* validation set different from the one used for evaluation, in contrast to the common practice in the literature where hyperparameter are directly selected on the evaluation dataset. All hyperparameters that yield the evaluation results reported in the paper are given in Appendix C.

ACKNOWLEDGMENTS

This project was funded by the CIFRE fellowship N°2020/1643 and supported in part by the AllegroAssai ANR project ANR-19-CHIA-0009. This work was granted access to the HPC resources of IDRIS under the allocation 2021-AD011012940 made by GENCI. Experiments presented in this paper were carried out using the Grid'5000 testbed, supported by a scientific interest group hosted by Inria and including CNRS, RENATER and several Universities as well as other organizations (see https://www.grid5000.fr).

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

# A  EXTENDED RELATED WORK

We further discuss some related works referenced in the main paper.

## A.1  REMINDERS ON KERNEL MEAN EMBEDDINGS

In this appendix we provide a high-level introduction to the notion of kernel mean embedding. We refer the reader to (Muandet et al., 2017) for a complete survey.

The idea of kernel mean embedding is to encode a probability distribution in an RKHS $\mathcal{H}$. Denoting $\mathcal{K} : \mathcal{X} \times \mathcal{X} \to \mathbb{R}$ the reproducing kernel of $\mathcal{H}$ defined on some space $\mathcal{X}$, the kernel mean embedding of a probability distribution $\mathbb{Q}$ defined on $\mathcal{X}$ is

$$\mu_{\mathbb{Q}} := \int_{\mathcal{X}} \mathcal{K}(\mathbf{u}, \cdot) d\mathbb{Q}(\mathbf{u}) \in \mathcal{H}. \tag{12}$$

In other words, the kernel mean embedding mapping $\mathbb{Q} \mapsto \mu_{\mathbb{Q}}$ transforms a probability distribution into an element in $\mathcal{H}$. As an application, this allows one to quantify the divergence between probabilities using the norm $\| \cdot \|_{\mathcal{H}}$ associated to $\mathcal{H}$. Given two probability distributions $\mathbb{Q}_1, \mathbb{Q}_2$ defined on $\mathcal{X}$, one can indeed quantify their divergence by

$$\|\mu_{\mathbb{Q}_1} - \mu_{\mathbb{Q}_2}\|_{\mathcal{H}} = \left\| \int_{\mathcal{X}} \mathcal{K}(\mathbf{u}, \cdot) d\mathbb{Q}_1(\mathbf{u}) - \int_{\mathcal{X}} \mathcal{K}(\mathbf{u}, \cdot) d\mathbb{Q}_2(\mathbf{u}) \right\|_{\mathcal{H}}, \tag{13}$$

which is precisely the MMD between $\mathbb{Q}_1$ and $\mathbb{Q}_2$ defined in (4).

## A.2  SAMPLE-CONTRASTIVE CRITERION

Given a batch of embeddings $\mathbf{Z}_I := \{\mathbf{z}_i\}_{i \in I}$ (that are not necessarily normalized), the general sample-constrative criterion of Garrido et al. (2023) is defined by:

$$\ell_c(\mathbf{Z}_I) = \sum_{i,i' \in I, \, i \neq i'} (\mathbf{z}_i^\top \mathbf{z}_{i'})^2. \tag{14}$$

Garrido et al. (2023) show that this criterion is minimized in many contrastive learning methods like (HaoChen et al., 2021). In the case where the embeddings are normalized, this sample-contrastive criterion can be derived from the proposed generic uniformity loss $\ell_u$ defined by (7) with the quadratic kernel $\mathcal{K}(\mathbf{u}, \mathbf{v}) = (\mathbf{u}^\top \mathbf{v})^2$ where $\mathbf{u}, \mathbf{v} \in \mathcal{S}^{q-1}$, as claimed in Section 1. Indeed, since $\|\mathbf{z}_i\|_2 = 1$ for all $i \in I$:

$$\ell_c(\mathbf{Z}_I) = \sum_{i,i' \in I} (\mathbf{z}_i^\top \mathbf{z}_{i'})^2 - \sum_{i \in I} (\mathbf{z}_i^\top \mathbf{z}_i)^2 = |I|^2 \ell_u(\mathbf{Z}_I) - |I|, \tag{15}$$

where $|I|$ is the batch size. Therefore, the sample-contrastive criterion in the normalized case is an estimator of the MMD associated to the quadratic kernel between the embedding distribution and the uniform distribution on the hypersphere.

## A.3  KERNEL DEPENDENCE MAXIMIZATION

We further explain the positioning of our paper with respect to (Li et al., 2021), which proposes a self-supervised learning method based on kernel dependence maximization, using the Hilbert-Schmidt Independence Criterion (HSIC) (Gretton et al., 2005). The HSIC measures the dependence between two random variables $X \in \mathcal{X}$ and $Y \in \mathcal{Y}$ using two RKHS $\mathcal{F}$ on $\mathcal{X}$ with kernel $k$ and $\mathcal{G}$ on $\mathcal{Y}$ with kernel $l$, in order to capture nonlinear correlations. It is defined as the squared MMD associated to the reproducing kernel of the tensor product space $\mathcal{F} \otimes \mathcal{G}$ between the joint probability distribution $\mathbb{P}_{X,Y}$ and the product $\mathbb{P}_X \mathbb{P}_Y$ of marginal probability distributions:

$$\text{HSIC}(X, Y) := \|\mu_{\mathbb{P}_{X,Y}} - \mu_{\mathbb{P}_X \mathbb{P}_Y}\|_{\mathcal{F} \otimes \mathcal{G}}^2, \tag{16}$$

where $\mathbb{Q} \mapsto \mu_{\mathbb{Q}}$ is the kernel mean embedding mapping defined by (12). Then, the self-supervised learning loss in (Li et al., 2021) is defined as:

$$\mathcal{L}_{\text{SSL-HSIC}} := -\text{HSIC}(Z, Y) + \gamma \sqrt{\text{HSIC}(Z, Z)}, \tag{17}$$

where $Z$ encodes embeddings of transformed images, and $Y$ encodes image identity as the index of the original image (before transformation) in the training dataset. By maximizing $\mathrm{HSIC}(Z, Y)$, the backbone learns image representations that are invariant to image transformations. To avoid collapse, high-variance representations are penalized by minimizing $\mathrm{HSIC}(Z, Z)$. This is similar to previous information-maximization methods (Bardes et al., 2022; Zbontar et al., 2021), with the difference that they take into account nonlinear correlations using kernels.

Although both our approach and the one in (Li et al., 2021) view self-supervised learning as a kernel method, we highlight here a main distinction between the two works. Both approaches use the MMD, but they do not use it to measure the same quantity. As explained above, Li et al. (2021) use the MMD to measure dependency between random variables (like $Z$ and $Y$), while the regularization loss we propose uses the MMD to measure the divergence between the embedding distribution and the uniform distribution on the hypersphere. As explained in Sections 1 and 2, this kernel approach for self-supervised learning is new in the literature and allows for the unification of several previous self-supervised learning methods as illustrated in Table 1.

Note that when the image identity $Y$ is a one-hot encoding, (Li et al., 2021) shows that

$$- \mathrm{HSIC}(Z, Y) = C \left( -\mathbb{E}_{(Z_1, Z_2) \sim \mathrm{pos}}[k(Z_1, Z_2)] + \mathbb{E}_{Z_3} \mathbb{E}_{Z_4}[k(Z_3, Z_4)] \right), \tag{18}$$

where $C > 0$ is a constant, $(Z_1, Z_2)$ is a positive pair of embeddings, i.e., they are embeddings of two transformations of the same original image, and $(Z_3, Z_4)$ is a pair of independent embeddings. In other words, $\mathrm{HSIC}(Z, Y)$ is proportional to the sum of an alignment term $-\mathbb{E}_{(Z_1, Z_2) \sim \mathrm{pos}}[k(Z, Z')]$ and an energy term $\mathbb{E}_{Z_3} \mathbb{E}_{Z_4}[k(Z_3, Z_4)]$, similarly to (3) combined with (7), which yields the proposed loss (2). Our paper shows that, if $k(\cdot, \cdot)$ is a rotation-invariant kernel on the hypersphere, then the energy term $\mathbb{E}_{Z_3} \mathbb{E}_{Z_4}[k(Z_3, Z_4)]$ is precisely the MMD between the embedding distribution and the uniform distribution on the hypersphere, cf. (6). However there are two differences between the maximization of $\mathrm{HSIC}(Z, Y)$ and the minimization of the proposed loss (2). First, the alignment term and the energy term in (18) are quantified with the *same* kernel $k(\cdot, \cdot)$, which is not the case in (2) where the alignment term is quantified by the $\ell_2$-distance between embeddings (equivalent to the linear kernel when the embeddings are normalized), and the uniformity term (7) is quantified by another rotation-invariant kernel. Second, the loss (2) is a *weighted* sum between the alignment loss (3) and the uniformity loss (7) controlled by the hyperparameter $\lambda$ that tunes the balance between the two terms, which is not the case of (18).

## B  THEORETICAL RESULTS

We provide proofs and more details about the theoretical results in the main text.

### B.1  PROOF OF LEMMA 2

Consider a rotation-invariant kernel $\mathcal{K}(\mathbf{u}, \mathbf{v})$ defined on the hypersphere $\mathcal{S}^{q-1}$ such that:

$$\mathcal{K}(\mathbf{u}, \mathbf{v}) = \sum_{\ell=0}^{+\infty} b_\ell P_\ell(q; \mathbf{u}^\top \mathbf{v}), \quad \forall \mathbf{u}, \mathbf{v} \in \mathcal{S}^{q-1}, \tag{19}$$

with weights $b_\ell \geq 0$ and $P_\ell(q; \cdot)$ the Legendre polynomial of order $\ell$ in dimension $q$. The proof of Lemma 2 relies on an orthonormal system of spherical harmonics. Let $\langle f, g \rangle_{(q)} := \int_{\mathcal{S}^{q-1}} fg \, d\sigma_{q-1}$ be the inner product in the space of continuous functions defined on $\mathcal{S}^{q-1}$ and, for each $\ell \in \mathbb{N}$, consider $\{Y_{\ell,k} \mid k = 1, \ldots, N(q, \ell)\}$ an orthonormal basis of spherical harmonics of order $\ell$ in dimension $q$ (homogeneous harmonic polynomials in $q$ variables restricted to $\mathcal{S}^{q-1}$, see e.g. (Müller, 2012) for more details), where $N(q, \ell)$ denotes the dimension of this space, which is by (Müller, 2012, Exercise 6, §3):

$$N(q, \ell) = \begin{cases} q & \text{for } \ell = 1, \\ \frac{(2\ell+q-2)(\ell+q-3)!}{\ell! \, (q-2)!} & \text{for } \ell \geq 2. \end{cases} \tag{20}$$

By the addition theorem (Müller, 2012, Theorem 2, §1):

$$\sum_{k=1}^{N(q,\ell)} Y_{\ell,k}(\mathbf{u}) Y_{\ell,k}(\mathbf{v}) = \frac{N(q, \ell)}{|\mathcal{S}^{q-1}|} P_\ell(q; \mathbf{u}^\top \mathbf{v}), \quad \mathbf{u}, \mathbf{v} \in \mathcal{S}^{q-1}. \tag{21}$$

Hence, the kernel $\mathcal{K}(\mathbf{u}, \mathbf{v})$ can be rewritten as:

$$\mathcal{K}(\mathbf{u}, \mathbf{v}) = \sum_{\ell=0}^{+\infty} \sum_{k=1}^{N(q,\ell)} \frac{b_\ell |\mathcal{S}^{q-1}|}{N(q,\ell)} Y_{\ell,k}(\mathbf{u}) Y_{\ell,k}(\mathbf{v}). \tag{22}$$

Since $\{Y_{\ell,k} \mid \ell = 0, \ldots, +\infty, k = 1, \ldots, N(q,\ell)\}$ is an orthonormal system for the inner product $\langle \cdot, \cdot \rangle_{(q)}$, and since $Y_{0,1}$ is constant on $\mathcal{S}^{q-1}$, we have for any integer $\ell$ and $k \in \{1, \ldots, N(q,\ell)\}$ that:

$$\int_{\mathcal{S}^{q-1}} Y_{\ell,k} d\sigma_{q-1} = \frac{1}{Y_{0,1}} \langle Y_{\ell,k}, Y_{0,1} \rangle_{(q)} = \begin{cases} \frac{1}{Y_{0,1}} & \text{if } \ell = 0, k = 1 \\ 0 & \text{otherwise} \end{cases}. \tag{23}$$

Moreover $Y_{0,1} = 1/\sqrt{|\mathcal{S}^{q-1}|}$, because $1 = \langle Y_{0,1}, Y_{0,1} \rangle_{(q)} = \int_{\mathcal{S}^{q-1}} Y_{0,1}^2 d\sigma_{q-1} = Y_{0,1}^2 |\mathcal{S}^{q-1}|$. Therefore, the kernel mean embedding of the uniform distribution on the hypersphere $\mathbb{U} := \sigma_{q-1}/|\mathcal{S}^{q-1}|$ associated to the kernel $\mathcal{K}$ is:

$$\begin{aligned}
\forall \mathbf{v} \in \mathcal{S}^{q-1}, \int_{\mathcal{S}^{q-1}} \mathcal{K}(\mathbf{u}, \mathbf{v}) d\mathbb{U}(\mathbf{u}) &= \int_{\mathcal{S}^{q-1}} \sum_{\ell=0}^{+\infty} b_\ell P_\ell(q; \mathbf{u}^\top \mathbf{v}) \frac{d\sigma_{q-1}(\mathbf{u})}{|\mathcal{S}^{q-1}|} \\
&= \sum_{\ell=0}^{+\infty} \int_{\mathcal{S}^{q-1}} \sum_{k=1}^{N(q,\ell)} \frac{b_\ell}{N(q,\ell)} Y_{\ell,k}(\mathbf{u}) Y_{\ell,k}(\mathbf{v}) d\sigma_{q-1}(\mathbf{u}) \\
&= \sum_{\ell=0}^{+\infty} \frac{b_\ell}{N(q,\ell)} \sum_{k=1}^{N(q,\ell)} \left[ \int_{\mathcal{S}^{q-1}} Y_{\ell,k}(\mathbf{u}) d\sigma_{q-1}(\mathbf{u}) \right] Y_{\ell,k}(\mathbf{v}) \\
&= b_0 \frac{1}{Y_{1,0}} Y_{1,0} = b_0,
\end{aligned} \tag{24}$$

where we inverted series and integral in the second equation using the dominated convergence theorem: the series $\sum_{\ell=0}^{+\infty} b_\ell P_\ell(q; \mathbf{u}^\top \mathbf{v})$ converges for every $\mathbf{u}, \mathbf{v}$, and for any $L$, $|\sum_{\ell=0}^{L} b_\ell P_\ell(q; \mathbf{u}^\top \mathbf{v})| \leq \sum_{\ell=0}^{L} |b_\ell P_\ell(q; \mathbf{u}^\top \mathbf{v})| \leq \sum_{\ell=0}^{+\infty} b_\ell = \sum_{\ell=0}^{+\infty} b_\ell P_\ell(q; 1)$, because $|P_\ell(q; \cdot)| \leq 1$ for all $\ell$ by (Müller, 2012, Lemma 2, §8), $P_\ell(q; 1) = 1$ for all $\ell$ by (Müller, 2012, §9), and $\sum_{\ell=0}^{+\infty} b_\ell P_\ell(q; 1) < +\infty$ is integrable on $\mathcal{S}^{q-1}$. This yields the first claim of Lemma 2.

Consider now any probability distribution $\mathbb{Q}$ defined on the hypersphere. The kernel mean embedding of $\mathbb{Q}$ is simply rewritten as:

$$\begin{aligned}
\forall \mathbf{v} \in \mathcal{S}^{q-1}, \int_{\mathcal{S}^{q-1}} \mathcal{K}(\mathbf{u}, \mathbf{v}) d\mathbb{Q}(\mathbf{u}) &= \int_{\mathcal{S}^{q-1}} \sum_{\ell=0}^{+\infty} b_\ell P_\ell(q; \mathbf{u}^\top \mathbf{v}) d\mathbb{Q}(\mathbf{u}) \\
&= b_0 \int_{\mathcal{S}^{q-1}} P_0(q; \mathbf{u}^\top \mathbf{v}) d\mathbb{Q}(\mathbf{u}) + \int_{\mathcal{S}^{q-1}} \sum_{\ell=1}^{+\infty} b_\ell P_\ell(q; \mathbf{u}^\top \mathbf{v}) d\mathbb{Q}(\mathbf{u}) \\
&= b_0 + \int_{\mathcal{S}^{q-1}} \tilde{\mathcal{K}}(\mathbf{u}, \mathbf{v}) d\mathbb{Q}(\mathbf{u}),
\end{aligned} \tag{25}$$

because the Legendre polynomial of order 0 is the constant function equal to 1 (see the closed form expression of $P_0(q; \cdot)$ in Theorem 1 in the main text) and $\int_{\mathcal{S}^{q-1}} d\mathbb{Q} = 1$. This ends the proof of Lemma 2.

### B.2 Legendre expansion of rotation-invariant kernels

We show that the kernel weights $b_\ell$ in the Legendre expansion (5) of the RBF kernel and the generalized distance kernel decay with a rate at least polynomial with respect to $\ell$.

**RBF kernel** The RBF kernel is defined as:

$$\mathcal{K}_{\text{RBF}}(\mathbf{u}, \mathbf{v}) = e^{-\sigma \|\mathbf{u}-\mathbf{v}\|_2^2} = e^{-2\sigma(1-\mathbf{u}^\top \mathbf{v})} \quad \text{for } \mathbf{u}, \mathbf{v} \in \mathcal{S}^q, \tag{26}$$

where $\sigma > 0$ is the scale of the RBF kernel. Denote $\varphi(t) := e^{-2\sigma(1-t)}$ for $t \in [-1, 1]$. Since the RBF kernel is positive definite and rotation-invariant, by Theorem 1, there exist weights $b_\ell \geq 0$, $\ell = 0, \ldots, +\infty$, such that:

$$\varphi(t) = e^{-2\sigma(1-t)} = \sum_{\ell=0}^{+\infty} b_\ell P_\ell(q; t) \quad \text{for } t \in [-1, 1]. \tag{27}$$

The Legendre polynomials $P_\ell(q; \cdot)$ are orthogonal on the interval $[-1, 1]$ with respect to the weight function $(1 - t^2)^{\frac{q-3}{2}}$, see e.g. (Müller, 2012):

$$\int_{-1}^{1} P_n(q; t) P_m(q; t)(1 - t^2)^{\frac{q-3}{2}} dt = 0 \quad \text{for } m \neq n. \tag{28}$$

Moreover, by (Müller, 2012, Exercise 3, §2):

$$\int_{-1}^{1} (P_n(q; t))^2 (1 - t^2)^{\frac{q-3}{2}} dt = \frac{|\mathcal{S}^{q-1}|}{|\mathcal{S}^{q-2}|} \frac{1}{N(q, n)} \quad \text{for any } n. \tag{29}$$

We multiply (27) by $P_\ell(q; t)(1 - t^2)^{\frac{q-3}{2}}$ and integrate the equation on $[-1, 1]$:

$$\begin{aligned}
\int_{-1}^{1} \varphi(t) P_\ell(q; t)(1 - t^2)^{\frac{q-3}{2}} dt &= \int_{-1}^{1} \sum_{n=0}^{+\infty} b_n P_n(q; t) P_\ell(q; t)(1 - t^2)^{\frac{q-3}{2}} dt \\
&= \sum_{n=0}^{+\infty} b_n \int_{-1}^{1} P_n(q; t) P_\ell(q; t)(1 - t^2)^{\frac{q-3}{2}} dt \\
&= b_\ell \frac{|\mathcal{S}^{q-1}|}{|\mathcal{S}^{q-2}|} \frac{1}{N(q, \ell)},
\end{aligned} \tag{30}$$

where the inversion between series and integral is justified by the dominated convergence theorem: the series $\sum_{n=0}^{+\infty} b_n P_n(q; t) P_\ell(q; t)(1 - t^2)^{\frac{q-3}{2}}$ converges for every $t$, and for any $N$, $|\sum_{n=0}^{N} b_n P_n(q; t) P_\ell(q; t)(1 - t^2)^{\frac{q-3}{2}}| \leq \sum_{n=0}^{+\infty} b_n (1 - t^2)^{\frac{q-3}{2}} := g(t)$ since $|P_n(q; \cdot)| \leq 1$ for any $n$ by (Müller, 2012, Lemma 2, §8), and $g$ is integrable on $[-1, 1]$ because $\sum_{n=0}^{+\infty} b_n < +\infty$. Hence:

$$b_\ell = N(q, \ell) \frac{|\mathcal{S}^{q-2}|}{|\mathcal{S}^{q-1}|} \int_{-1}^{1} \varphi(t) P_\ell(q; t)(1 - t^2)^{\frac{q-3}{2}} dt. \tag{31}$$

By the Rodrigues rule (Müller, 2012, Exercise 1, §2), since $\varphi$ has continuous derivatives of all orders on $[-1, 1]$, we have:

$$b_\ell = N(q, \ell) \frac{|\mathcal{S}^{q-2}|}{|\mathcal{S}^{q-1}|} \frac{\Gamma(\frac{q-1}{2})}{2^\ell \Gamma(\ell + \frac{q-1}{2})} \int_{-1}^{1} \varphi^{(\ell)}(t)(1 - t^2)^{\ell + \frac{q-3}{2}} dt, \quad \ell \in \mathbb{N}, \tag{32}$$

where $\varphi^{(\ell)}$ is the $\ell$-th derivative of $\varphi$, which is $\varphi^{(\ell)}(t) = e^{-2\sigma}(2\sigma)^\ell e^{2\sigma t}$. We now show that the weights $b_\ell$ decay very fast with respect to $\ell$. We bound the integral:

$$\int_{-1}^{1} \varphi^{(\ell)}(t)(1 - t^2)^{\ell + \frac{q-3}{2}} dt = \int_{-1}^{1} e^{-2\sigma}(2\sigma)^\ell e^{2\sigma t}(1 - t^2)^{\ell + \frac{q-3}{2}} dt \leq \int_{-1}^{1} (2\sigma)^\ell dt = 2(2\sigma)^\ell. \tag{33}$$

Hence:

$$b_\ell \leq 2 N(q, \ell) \frac{|\mathcal{S}^{q-2}|}{|\mathcal{S}^{q-1}|} \frac{\Gamma(\frac{q-1}{2})}{2^\ell \Gamma(\ell + \frac{q-1}{2})} (2\sigma)^\ell. \tag{34}$$

Denote $(a)_n := \Gamma(n + a)/\Gamma(a)$ the Pochhammer symbol defined for any integer $n$ and any scalar $a$. By the Stirling approximation of the Gamma function (Spiegel et al., 2013, (25.15)), the asymptotic behavior of $(a)_n$ when $n$ goes to infinity is:

$$(a)_n \sim \frac{\sqrt{2\pi}}{\Gamma(a)} e^{-n} n^{a+n-1/2} \quad \text{as } n \to \infty. \tag{35}$$

Moreover, for a fixed dimension $q$, the asymptotic behavior of $N(q, \ell)$ defined by (20) when $\ell$ goes to infinity is:

$$N(q, \ell) \sim \frac{2}{(q-2)!} \ell^{q-2} \quad \text{as } \ell \to \infty. \tag{36}$$

Therefore, the asymptotic behavior of $b_\ell$ as $\ell$ goes to infinity is:

$$b_\ell = \mathcal{O}\left(\sigma^\ell e^\ell \ell^{q/2-1-\ell}\right) \quad \text{as } \ell \to +\infty. \tag{37}$$

**Generalized distance kernel**  For $\frac{q-1}{2} < s < \frac{q+1}{2}$, the generalized distance kernel on the hypersphere $\mathcal{S}^{q-1}$ is defined in (Brauchart et al., 2014, Section 5) as:

$$\mathcal{K}_{\text{gd}}^{(s)}(\mathbf{u}, \mathbf{v}) := 2V_{q-1-2s}(\mathcal{S}^{q-1}) - \|\mathbf{u} - \mathbf{v}\|_2^{2s-q+1} \quad \text{for } \mathbf{u}, \mathbf{v} \in \mathcal{S}^{q-1}, \tag{38}$$

where

$$V_{q-1-2s}(\mathcal{S}^{q-1}) := \int_{\mathcal{S}^{q-1}} \int_{\mathcal{S}^{q-1}} \|\mathbf{u} - \mathbf{v}\|_2^{2s-q+1} d\sigma_{q-1}(\mathbf{u}) d\sigma_{q-1}(\mathbf{v}) = 2^{2s-1} \frac{\Gamma(q/2)\Gamma(s)}{\sqrt{\pi}\Gamma((q-1)/2+s)}. \tag{39}$$

Following Brauchart et al. (2014, Section 5), the Legendre expansion of the generalized distance kernel $\mathcal{K}_{\text{gd}}^{(s)}$ is:

$$\mathcal{K}_{\text{gd}}^{(s)}(\mathbf{u}, \mathbf{v}) = V_{q-1-2s}(\mathcal{S}^{q-1}) + \sum_{\ell=1}^{+\infty} \alpha_\ell^{(s)} N(q, \ell) P_\ell(q; \mathbf{u}^\top \mathbf{v}), \tag{40}$$

$$\alpha_\ell^{(s)} := -V_{q-1-2s}(\mathcal{S}^{q-1}) \frac{((q-1)/2 - s)_\ell}{((q-1)/2 + s)_\ell}, \quad \ell \geq 1. \tag{41}$$

The kernel weights indeed decay polynomially with respect to $\ell$, because according to Brauchart et al. (2014, Section 5), the asymptotic behavior of the $\alpha_\ell^{(s)}$ is:

$$\alpha_\ell^{(s)} \sim 2^{2s-1} \frac{\Gamma(q/2)\Gamma(s)}{\sqrt{\pi}\Gamma((q-1)/2 - s)} \ell^{-2s} \quad \text{as } \ell \to +\infty. \tag{42}$$

### B.3 Connection between SFRIK and VICReg

Consider a rotation-invariant kernel $\tilde{\mathcal{K}}(\mathbf{u}, \mathbf{v}) := \sum_{\ell=1}^{+\infty} b_\ell P_\ell(q; \mathbf{u}^\top \mathbf{v})$ defined on $\mathcal{S}^{q-1}$ such that $b_\ell \geq 0$ for $\ell \in \{1, \dots, +\infty\}$, with $b_1, b_2 > 0$. To show the connection between SFRIK and VICReg, we construct a high-dimensional feature map $\Phi : \mathcal{S}^{q-1} \to \ell_2(\mathbb{N})$, where $\ell_2(\mathbb{N})$ denotes the space of square-summable sequences with its canonical inner product $\langle \cdot, \cdot \rangle_{\ell_2}$, such that $\langle \Phi(\mathbf{u}), \Phi(\mathbf{v}) \rangle_{\ell_2} = \tilde{\mathcal{K}}(\mathbf{u}, \mathbf{v})$ for any $\mathbf{u}, \mathbf{v} \in \mathcal{S}^{q-1}$.

One way to construct such a feature map is to consider an orthonormal system of spherical harmonics. For any integer $\ell$, denote $\{Y_{\ell,k}\}_{k=1}^{N(q,\ell)}$ an orthonormal basis of spherical harmonics of order $\ell$ in dimension $q$. By the addition theorem (Müller, 2012, Theorem 2, §1) recalled in (21), the kernel $\tilde{\mathcal{K}}(\mathbf{u}, \mathbf{v})$ admits the decomposition:

$$\begin{aligned}
\tilde{\mathcal{K}}(\mathbf{u}, \mathbf{v}) = \sum_{\ell=1}^{+\infty} b_\ell P_\ell(q; \mathbf{u}^\top \mathbf{v}) &= \sum_{\ell=1}^{+\infty} \sum_{k=1}^{N(q,\ell)} \frac{b_\ell |\mathcal{S}^{q-1}|}{N(q,\ell)} Y_{\ell,k}(\mathbf{u}) Y_{\ell,k}(\mathbf{v}) \\
&= \langle \Phi(\mathbf{u}), \Phi(\mathbf{v}) \rangle_{\ell_2},
\end{aligned} \tag{43}$$

where

$$\Phi := \left(\sqrt{\frac{b_\ell |\mathcal{S}^{q-1}|}{N(q,\ell)}} \Phi_\ell\right)_{\ell=1}^{+\infty} \quad \text{with} \quad \Phi_\ell : \begin{cases} \mathcal{S}^{q-1} \to \mathbb{R}^{N(q,\ell)} \\ \mathbf{u} \mapsto (Y_{\ell,k}(\mathbf{u}))_{k=1}^{N(q,\ell)} \end{cases} \quad \text{for } \ell \in \{1, \dots, +\infty\}. \tag{44}$$

Then, the MMD in (6) between any probability distribution $\mathbb{Q}$ defined on the hypersphere and the uniform distribution $\mathbb{U}$ on the hypersphere can be written as the norm in $\ell_2(\mathbb{N})$ of the generalized

moment of $\mathbb{Q}$ with the mapping $\Phi$:

$$
\begin{aligned}
\mathrm{MMD}^2(\mathbb{Q}, \mathbb{U}) &= \mathbb{E}_{\mathbf{z}, \mathbf{z}' \sim \mathbb{Q}}[\tilde{\mathcal{K}}(\mathbf{z}, \mathbf{z}')] \\
&= \mathbb{E}_{\mathbf{z}, \mathbf{z}' \sim \mathbb{Q}}[\langle \Phi(\mathbf{z}), \Phi(\mathbf{z}') \rangle_{\ell_2}] \\
&= \langle \mathbb{E}_{\mathbf{z} \sim \mathbb{Q}}[\Phi(\mathbf{z})], \mathbb{E}_{\mathbf{z}' \sim \mathbb{Q}}[\Phi(\mathbf{z}')] \rangle_{\ell_2} = \left\| \mathbb{E}_{\mathbf{z} \sim \mathbb{Q}}[\Phi(\mathbf{z})] \right\|_{\ell_2}^2 \\
&= \sum_{\ell=1}^{+\infty} \frac{b_\ell |\mathcal{S}^{q-1}|}{N(q, \ell)} \left\| \mathbb{E}_{\mathbf{z} \sim \mathbb{Q}}[\Phi_\ell(\mathbf{z})] \right\|_2^2 .
\end{aligned}
\tag{45}
$$

We now explain how to construct explicitly an orthonormal basis of spherical harmonics $\{Y_{\ell,k} \mid \ell = 1, \ldots, +\infty;\ k = 1, \ldots, N(q, \ell)\}$, based on the following theorem.

**Theorem 3 (Axler et al. (2013, Theorem 5.25))** *For any order $\ell \in \mathbb{N}$ and any dimension $q \geq 3$, the family*

$$
\{Y'_{\ell,k}\}_{k=1}^{N(q,\ell)} := \left\{ \mathbf{u} \mapsto \partial_1^{\alpha_1} \partial_2^{\alpha_2} \ldots \partial_q^{\alpha_q} \|\mathbf{u}\|_2^{2-q} \mid \alpha_1 + \alpha_2 + \ldots + \alpha_q = \ell \text{ and } \alpha_1 \leq 1 \right\}
\tag{46}
$$

*is a (non-orthonormal) basis of the space of spherical harmonics of order $\ell$ in dimension $q$, where $\alpha_j$ ($j = 1, ..., q$) are nonnegative integers, and $\partial_j^{\alpha_j}$ denotes the $\alpha_j$-th partial derivative with respect to the $j$-th coordinate.*

Typically, we construct the orthonormal basis $\{Y_{\ell,k}\}_{k=1}^{N(q,\ell)}$ by orthonormalizing the basis $\{Y'_{\ell,k}\}_{k=1}^{N(q,\ell)}$ of Theorem 3 using, e.g., the Gram-Schmidt procedure. For $\ell = 1, \ldots, +\infty$, denote:

$$
\Phi'_\ell : \ \mathcal{S}^{q-1} \to \mathbb{R}^{N(q,\ell)}, \ \mathbf{u} \mapsto (Y'_{\ell,k}(\mathbf{u}))_{k=1}^{N(q,\ell)}.
\tag{47}
$$

Then, for each $\ell = 1, \ldots, +\infty$, there exists a lower triangular matrix $\mathbf{M}_\ell$ such that:

$$
\Phi_\ell(\mathbf{u}) = \mathbf{M}_\ell \Phi'_\ell(\mathbf{u}), \quad \text{for all } \mathbf{u} \in \mathcal{S}^{q-1}.
\tag{48}
$$

Remark that it is possible to compute explicitly the entries of the matrices $\mathbf{M}_\ell$, $\ell = 1, \ldots, +\infty$, because there exists a closed-form expression for the inner product $\langle Y'_{\ell,k}, Y'_{\ell,k'} \rangle_{(q)}$ for any $\ell, k, k'$: indeed, the function $Y'_{\ell,k}$ for any $\ell, k$ is a polynomial defined on the hypersphere, and the integral of any monomial with respect to the measure $\sigma_{q-1}$ on the hypersphere $\mathcal{S}^{q-1}$ admits a closed-form expression given by Weyl (1939, Section 3).

By injecting (48) in (45), we obtain:

$$
\mathrm{MMD}^2(\mathbb{Q}, \mathbb{U}) = \sum_{\ell=1}^{+\infty} \frac{b_\ell |\mathcal{S}^{q-1}|}{N(q, \ell)} \left\| \mathbf{M}_\ell \mathbb{E}_{\mathbf{z} \sim \mathbb{Q}}[\Phi'_\ell(\mathbf{z})] \right\|_2^2 .
\tag{49}
$$

This yields the claim of Section 3.3 by remarking with Theorem 3 that the families

$$
\begin{aligned}
\{Y'_{1,k}\}_{k=1}^{N(q,1)} &= \left\{ \mathbf{u} \mapsto u^j \mid 1 \leq j \leq q \right\}, \\
\{Y'_{2,k}\}_{k=1}^{N(q,2)} &= \left\{ \mathbf{u} \mapsto u^j u^{j'} \mid 1 \leq j < j' \leq q \right\} \cup \left\{ \mathbf{u} \mapsto (u^j)^2 - \frac{1}{q} \mid 2 \leq j \leq q \right\}
\end{aligned}
\tag{50}
$$

are bases of the space of spherical harmonics of order 1 and 2 in dimension $q$.

## B.4 REGULARIZATION LOSS OF SIMCLR, AUH AND VICREG

In SimCLR, AUH, VICReg and SFRIK, each image $\mathbf{x}_i$ in a batch $\{\mathbf{x}_i\}_{i \in I}$ is augmented into two different views $\mathbf{x}_i^{(1)}$ and $\mathbf{x}_i^{(2)}$, which are encoded into two embeddings $\mathbf{z}_i^{(1)}$ and $\mathbf{z}_i^{(2)}$. These embeddings are normalized in SimCLR, AUH and SFRIK, but not in VICReg. This yields two batches of embeddings $\mathbf{Z}_I^{(v)} := \{\mathbf{z}_i^{(v)}\}_{i \in I}$ ($v = 1, 2$). The four methods share the same form of loss function:

$$
\ell(\mathbf{Z}_I^{(1)}, \mathbf{Z}_I^{(2)}) := \lambda\, \ell_a(\mathbf{Z}_I^{(1)}, \mathbf{Z}_I^{(2)}) + \mu\, \ell_r(\mathbf{Z}_I^{(1)}, \mathbf{Z}_I^{(2)}),
\tag{51}
$$

for some scalars $\lambda, \mu > 0$, where $\ell_a$ is the alignment loss defined by (3) (which is the same for all the four methods), and $\ell_r$ is the regularization loss specific to each method.

**SimCLR**    The regularization loss in SimCLR is:

$$\ell_r(\mathbf{Z}_I^{(1)}, \mathbf{Z}_I^{(2)}) = \frac{1}{2|I|} \sum_{v=1}^{2} \sum_{i \in I} \log \left( \sum_{v'=1}^{2} \sum_{i' \in I} \mathbb{1}_{[(v,i) \neq (v',i')]} \exp \left( \mathbf{z}_i^{(v)\top} \mathbf{z}_{i'}^{(v')}/\tau \right) \right), \quad (52)$$

where $\tau > 0$ is a hyperparameter of the method called the temperature, and $\mathbb{1}_{[(v,i) \neq (v',i')]}$ is equal to 1 if $(v,i) \neq (v',i')$, and 0 otherwise. The scalars $\lambda, \mu$ are fixed at $\lambda = \frac{1}{\tau}$ and $\mu = 1$.

**Alignment & Uniformity**    The regularization loss in AUH is:

$$\ell_r(\mathbf{Z}_I^{(1)}, \mathbf{Z}_I^{(2)}) = \frac{1}{2|I|^2} \sum_{v=1}^{2} \log \left( \sum_{i \in I} \sum_{i' \in I} \exp(-t\|\mathbf{z}_i^{(v)} - \mathbf{z}_{i'}^{(v)}\|_2^2) \right), \quad (53)$$

where $t > 0$ is a hyperparameter called the scale of the RBF kernel. The scalar $\lambda$ is tuned as a hyperparameter and $\mu$ is fixed at $\mu = 1$.

**VICReg**    As introduced in Section 3.3, the regularization loss in VICReg is:

$$\ell_r(\mathbf{Z}_I^{(1)}, \mathbf{Z}_I^{(2)}) = \frac{1}{2} \left[ v(\mathbf{Z}_I^{(1)}) + v(\mathbf{Z}_I^{(2)}) \right] + \frac{1}{2\mu} \left[ c(\mathbf{Z}_I^{(1)}) + c(\mathbf{Z}_I^{(2)}) \right], \quad (54)$$

where $\mu$ is the scalar from (51). Here, $v(\cdot)$ and $c(\cdot)$ are respectively the variance and covariance terms defined by (9). Both $\lambda$ and $\mu$ are tuned as hyperparameters.

## C    EXPERIMENTAL SETTING

In the interest of reproducible research, we give more details about the setting of our experiments presented in Section 4.

### C.1    IN20% DATASET DESCRIPTION

The datasets used in our experiments include a subset of 20% of ImageNet-1000 as in (Gidaris et al., 2021). This reduced dataset, denoted IN20%, contains all the 1000 classes of ImageNet, but we keep only 260 images per class. The 260 images extracted are the same as those extracted in the official implementation of OBoW (https://github.com/valeoai/obow). In Section 4.2, we also use *another* 20% subset of the ImageNet train set as a separate validation set for hyperparameter tuning (see Appendix C.4 below). The construction of this validation set is based on the code of OBoW, and will be exactly detailed in our code that will be released at publication.

### C.2    IMAGE AUGMENTATIONS

We follow the same image augmentation pipeline as in (Bardes et al., 2022). Our experiments include the following image augmentations implemented by PyTorch (`torchvision.transforms`):

- `RandomResizedCrop(224, scale=(0.08, 1.0))`: crop a random area of the image between 8% and 100% of the total area, and resize it to an image of size $224 \times 224$;
- `RandomHorizontalFlip()`: flip horizontally an image;
- `ColorJitter(brightness=0.4, contrast=0.4, saturation=0.2, hue=0.1)`: randomly change brightness, contrast, saturation and hue of an image by a factor randomly sampled in respectively $[0.6, 1.4]$, $[0.6, 1.4]$, $[0.8, 1.2]$ and $[-0.1, 0.1]$.
- `RandomGrayscale()`: convert an image into grayscale.

We also use image augmentations implemented by PIL, as in VICReg's code available at https://github.com/facebookresearch/vicreg:

- `GaussianBlur()`: blur an image using a Gaussian kernel with standard deviation uniformly sampled in $[0.1, 2.0]$;
- `Solarization()`: randomly invert all pixel values above a threshold, which is 130.

In our experiments, the first image view is obtained by composing the following random augmentations: random cropping resized to $224 \times 224$, random horizontal flip applied with probability 0.5, random color jittering applied with probability 0.8, random grayscale conversion applied with probability 0.2, random Gaussian blur applied with probability 0.1, and random solarization applied with probability 0.2. The second view is obtained by composing the same random augmentations as the first view, except that Gaussian blur is applied every time (probability 1), and solarization is never applied (probability 0).

### C.3 EVALUATION PROTOCOL

We describe the downstream tasks on which self-supervision methods are evaluated in our experiments of Section 4.

**Linear probing on IN20% and IN100%**  Following, e.g., (Bardes et al., 2022), the weights of the backbone (ResNet-18 or ResNet-50) are frozen and a linear layer followed by a softmax on top of the backbone is trained in a supervised setting on a training set. Then the model is evaluated on a test set. The training set is either IN20% or IN100%, but with labels. The test set is the validation set of ImageNet. The linear layer is trained using an SGD optimizer with momentum parameter equal to 0.9 during 100 epochs. We apply a weight decay of $10^{-6}$. The learning rate follows a cosine decay scheduling. The batch size is fixed at 256. Training images are augmented by composing a random cropping of an area between 8% and 100% of the total area resized to $224 \times 224$, and a random horizontal flip of probability 0.5. Images at test time are resized to $256 \times 256$, and cropped at the center with a size $224 \times 224$. The initial learning rate is tuned as a hyperparameter, and we report the top-1 accuracy on the validation set of ImageNet obtained after the last training epoch, along with the corresponding top-5 accuracy. The code that we use for linear probing on IN20% or IN100% is adapted from (Bardes et al., 2022) available at `https://github.com/facebookresearch/vicreg`.

**Linear probing on Places205**  We use the code of Gidaris et al. (2021), available at `https://github.com/valeoai/obow`, for the evaluation by linear probing on Places205. The weights of the backbone (ResNet-50) pretrained on IN100% are frozen and a linear prediction layer is trained for the classification task on Places205. We note that a batch normalization layer with *non*-learnable scale and bias parameters is added at the output of the backbone in (Gidaris et al., 2021). The linear prediction layer is trained with an SGD optimizer with a 0.9 momentum parameter during 28 epochs. The weight decay is $10^{-4}$. The batch size is 256. The learning rate decreases by a factor of 10 at epoch 10 and epoch 20. We use the same image augmentation pipeline for training and testing as in linear probing on IN100%. The initial learning rate is tuned as a hyperparameter, and we report the top-1 accuracy on the validation set of Places205 obtained after the last training epoch, along with the corresponding top-5 accuracy.

**Linear classification on VOC2007**  After pretraining a ResNet backbone, we use the VISSL library (Goyal et al., 2021) to extract features of VOC2007 images resized to $224 \times 224$ by taking the output of the last average pooling layer of the pretrained ResNet backbone. We then learn a linear SVM with LIBLINEAR (Fan et al., 2008) on top of these features to predict the presence or the absence of a given class in the test images. An average precision score is then computed for each class after a 3-fold cross-validation, and we report the mean score over all classes as the mean average precision (mAP).

**Semi-supervised learning**  After pretraining a ResNet backbone by self-supervision, we fine-tune this backbone and the linear classifier on the ImageNet classification task with only 1% or 10% of the labeled data. The labeled images that are considered in these subsets are the ones used in the official code of SimCLR available at `https://github.com/google-research/simclr`. We use an SGD optimizer with momentum parameter equal to 0.9 during 20 epochs, without weight decay. The batch size is fixed at 256. The learning rates of the backbone and the linear classifier follow a cosine decay scheduling with different initial learning rates. These initial learning rates are tuned as hyperparameters. We report the top-1 accuracy on the validation set of ImageNet obtained after the last training epoch, along with the corresponding top-5 accuracy. We use the same image augmentation pipeline for training and testing as in linear probing on IN100%. The code that we use for semi-supervised learning with few labels of IN100% is the one of Bardes et al. (2022) available at `https://github.com/facebookresearch/vicreg`.

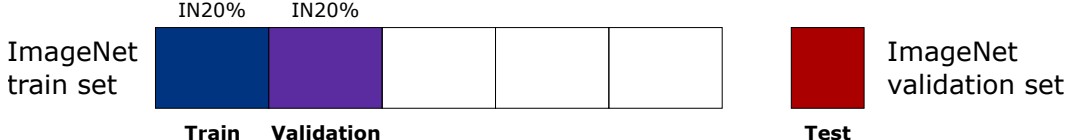

Figure 2: ImageNet dataset split for hyperparameter tuning in experiments of Section 4.2 on IN20%, as explained in Appendix C.4

**Weighted kNN classification** We follow the usual protocol of Wu et al. (2018); Caron et al. (2021). We compute the normalized representations $f_\theta(\mathbf{x}_i)$ of the images $\mathbf{x}_i$, $i \in [N]$, in the training set. The label of an image $\mathbf{x}_{\text{test}}$ in the test set is predicted by a weighted vote of its $k$ nearest neighbors $\mathcal{N}_k$ in the representation space: the class $c$ gets a score of $w_c :=$ $\sum_{i \in \mathcal{N}_k} \exp(f_\theta(\mathbf{x}_i)^\top f_\theta(\mathbf{x}_{\text{test}})/0.07)\mathbb{1}_{[c_i=c]}$ where $f_\theta(\mathbf{x}_{\text{test}})$ is normalized, $c_i$ is the class of $\mathbf{x}_i$, and $\mathbb{1}_{[c_i=c]}$ is equal to 1 if $c_i = c$, and 0 otherwise. We report the kNN classification top-1 accuracy for $k = 20$. The image augmentation pipeline for both training and testing is the following one: images are resized to $256 \times 256$, and cropped at the center with a size $224 \times 224$. The code that we use for kNN classification is the one of Caron et al. (2021) available at https://github.com/facebookresearch/dino.

## C.4 HYPERPARAMETERS FOR EXPERIMENTS ON IN20%

We describe in detail our hyperparameter tuning protocol for the experiments in Section 4.2 on IN20%. For a rigorous tuning, it is important that the dataset used for the final evaluation remains unseen during pretraining and hyperparameter tuning. For each pretraining method, we pretrain on the IN20% training set (blue subset in Figure 2) a backbone for each choice of hyperparameters. These backbones are then evaluated by weighted kNN classification on a separate validation set, which is another 20% subset of the ImageNet train set (purple subset in Figure 2), and we select the hyperparameters yielding the highest top-1 accuracy on this kNN evaluation. Then, we tune the learning rate for the linear probing evaluation, again on our separate validation set (purple subset in Figure 2). Finally, we use the model trained with the best learning rate discovered for linear probing evaluation on the usual ImageNet validation set (red subset in Figure 2), which has never been seen during hyperparameter tuning.

We report the values of the optimal hyperparameters found after our hyperparameter tuning on a separate validation set, for each pretraining experiment on IN20% with a ResNet-18 presented in Section 4.2. These hyperparameters yield the evaluation results reported in Section 4.2 for linear probing on the usual ImageNet validation set. We recall that the hyperparameters specific to each self-supervision method was introduced in Appendix B.4.

**SimCLR.** For each embedding dimension, we fix the batch size at 2048, and tune the temperature $\tau$ and the base learning rate $base\_lr$ for pretraining with SimCLR. Then, we tune the initial learning rate $lr\_head$ for linear probing on IN20%. The optimal hyperparameters are shown in Table 6.

**AUH.** For each embedding dimension, we fix the batch size at 2048, and tune the alignment weight $\lambda$, the scale of the RBF kernel $t$, and the base learning rate $base\_lr$ for pretraining with AUH. Then we tune the initial learning rate $lr\_head$ for linear probing on IN20%. The optimal hyperparameters are shown in Table 7.

Table 6: Hyperparameter choice for SimCLR pretrained on IN20% with a ResNet-18 during 100 epochs, evaluated by linear probing on IN20%.

| Dimension | Temperature | $base\_lr$ | $lr\_head$ |
|-----------|-------------|------------|------------|
| $q = 1024$ | 0.15 | 1.0 | 0.2 |
| $q = 2048$ | 0.15 | 1.0 | 0.2 |
| $q = 4096$ | 0.15 | 1.0 | 0.2 |
| $q = 8192$ | 0.15 | 0.8 | 0.2 |

Table 7: Hyperparameter choice for AUH pretrained on IN20% with a ResNet-18 during 100 epochs, evaluated by linear probing on IN20%.

| Dimension | Alignment weight $\lambda$ | Scale $t$ | $base\_lr$ | $lr\_head$ |
|---|---|---|---|---|
| $q = 1024$ | 400 | 2.5 | 1.0 | 10.0 |
| $q = 2048$ | 1000 | 2.5 | 1.0 | 4.0 |
| $q = 4096$ | 2000 | 2.5 | 1.0 | 1.0 |
| $q = 8192$ | 3000 | 2.5 | 1.0 | 2.0 |

Table 8: Hyperparameter choice for VICReg pretrained on IN20% with a ResNet-18 during 100 epochs, evaluated by linear probing on IN20%.

| Dimension | Alignment weight $\lambda$ | Variance weight $\mu$ | $base\_lr$ | $lr\_head$ |
|---|---|---|---|---|
| $q = 1024$ | 4 | 10 | 0.4 | 0.2 |
| $q = 2048$ | 4 | 4 | 0.7 | 1.0 |
| $q = 4096$ | 10 | 10 | 0.6 | 0.2 |
| $q = 8192$ | 10 | 10 | 0.7 | 0.2 |

Table 9: Hyperparameter choice for SFRIK pretrained on IN20% with a ResNet-18 during 100 epochs, evaluated by linear probing on IN20%.

| Order | Dimension | Alignment weight $\lambda$ | Kernel weights $(b_1, b_2, b_3)$ | $base\_lr$ | $lr\_head$ |
|---|---|---|---|---|---|
| $L = 1$ | $q = 8192$ | 10000 | $(1, 0, 0)$ | 0.4 | 10.0 |
| $L = 2$ | $q = 1024$ | 400 | $(1, 40, 0)$ | 1.0 | 2.0 |
| | $q = 2048$ | 400 | $(1, 40, 0)$ | 1.0 | 4.0 |
| | $q = 4096$ | 1000 | $(1, 40, 0)$ | 1.0 | 4.0 |
| | $q = 8192$ | 2000 | $(1, 20, 0)$ | 1.0 | 10.0 |
| $L = 3$ | $q = 8192$ | 4000 | $(1, 40, 40)$ | 1.2 | 1.0 |

**VICReg.** For each embedding dimension, we fix the batch size at 2048, and we tune the alignment weight $\lambda$, the variance weight $\mu$, and the base learning rate $base\_lr$ for pretraining with VICReg. Then, we tune the initial learning rate $lr\_head$ for linear probing on IN20%. The optimal hyperparameters are shown in Table 8.

**SFRIK, batch size 2048.** For each embedding dimension, we fix the batch size at 2048, and tune the alignment weight $\lambda$ in (2), the kernel weights $b_\ell$ ($\ell \in \{2, 3\}$) in (8), and the base learning rate $base\_lr$ for pretraining with SFRIK. Without loss of generality, the first kernel weight $b_1$ in (8) is fixed at $b_1 = 1$. Then, we tune the initial learning rate $lr\_head$ for linear probing on IN20%. The optimal hyperparameters are shown in Table 9.

**SFRIK, batch size 4096.** We fix the dimension at $q = 8192$, and the batch size at 4096, and tune the alignment weight $\lambda$ in (2), the kernel weights $b_\ell$ ($\ell \in \{2, 3\}$) in (8), and the base learning rate $base\_lr$ for pretraining with SFRIK. Without loss of generality, the first kernel weight $b_1$ in (8) is fixed at $b_1 = 1$. Then, we tune the initial learning rate $lr\_head$ for linear probing on IN20%. The optimal hyperparameters are: $\lambda = 4000$, $(b_1, b_2, b_3) = (1, 20, 0)$, $base\_lr = 0.8$, $lr\_head = 1.0$. This yields a top-1 accuracy of 46.3 for linear probing on IN20%, meaning that it is not necessary to use a batch size larger than 2048 in SFRIK to obtain a better performance.

## C.5 HYPERPARAMETERS FOR THE ABLATION ON THE KERNEL CHOICE

We detail the experimental setting of our ablation study on the kernel choice for the generic uniformity term (7) in Table 3 of Section 4.2. We follow the protocol of Section 4.2, with an embedding dimension of $q = 8192$. The training loss is $\lambda\ell_a(\mathbf{Z}_I^{(1)}, \mathbf{Z}_I^{(2)}) + 0.5(\ell_u(\mathbf{Z}_I^{(1)}) + \ell_u(\mathbf{Z}_I^{(2)}))$, where $\ell_u$ is the loss (7) with an RBF or a general distance kernel.

**RBF kernel.** The kernel is $\mathcal{K}(\mathbf{u}, \mathbf{v}) = e^{-t\|\mathbf{u}-\mathbf{v}\|_2^2}$. We fix the batch size at 2048, and tune the alignment weight $\lambda$, the scale of the RBF kernel $t$, and the base learning rate $base\_lr$ for pretraining. Then, we tune the initial learning rate $lr\_head$ for linear probing on IN20%. The optimal hyperparameters are: $\lambda = 100$, $t = 2$, $base\_lr = 1.0$, $lr\_head = 10$.

**Generalized distance kernel.** The kernel is $\mathcal{K}(\mathbf{u}, \mathbf{v}) = C - \|\mathbf{u} - \mathbf{v}\|_2^p$, where we fixed $C = 0$ because the value of this constant does not change the gradients of the optimization problem, and $p = 2$ because we verified empirically that choosing $p < 2$ yields poor results. We fix the batch size at 2048, and tune the alignment weight $\lambda$, and the base learning rate $base\_lr$ for pretraining. Then, we tune the initial learning rate $lr\_head$ for linear probing on IN20%. The optimal hyperparameters are: $\lambda = 10000$, $base\_lr = 0.6$, $lr\_head = 10$.

As shown in Table 3, a truncated kernel is empirically a better choice than the RBF or the generalized distance kernel for the uniformity term (7).

## C.6 Hyperparameters for experiments on IN100%

Table 10 reports the selected hyperparameters for pretraining ResNet-50 on IN100% with SFRIK in Section 4.3. The tuning protocol is as follows. Since hyperparameter tuning is costly on IN100%, we pretrain several ResNet-50 for different values of kernel weights $b_\ell$, alignment weight $\lambda$ and base learning rate $base\_lr$, and pause the pretraining after 50 epochs. We evaluate the obtained backbones on kNN classification (top-1 accuracy, $k = 20$), and select the best performing backbones. Then we continue pretraining these selected backbones until reaching epoch 200 or 400. Finally we select the hyperparameters that yield the highest top-1 accuracy for linear probing on IN100% after 200 or 400 epochs of pretraining. Because of the conclusions of Appendix D.1, our hyperparameter tuning follows the common practice in the self-supervised learning literature where hyperparameters are selected by measuring the performance on the validation set of ImageNet. Note that we verified experimentally a posteriori that the hyperparameters obtained by our tuning protocol are similar to the ones obtained from a tuning on a smaller dataset like STL-10 (Coates et al., 2011). This means that an alternative is to tune the hyperparameters on STL-10, and generalize these hyperparameters to pretrain SFRIK on IN100%.

Table 10: Hyperparameter choice for SFRIK pretrained on IN100% with a ResNet-50 during 200 or 400 epochs.

| Dimension | Order | Epoch | Alignment weight $\lambda$ | Kernel weights $(b_1, b_2, b_3)$ | $base\_lr$ |
|---|---|---|---|---|---|
| $q = 8192$ | $L = 2$ | 200 | 4000 | $(1, 20, 0)$ | 0.4 |
| | $L = 3$ | 200 | 4000 | $(1, 40, 40)$ | 0.4 |
| $q = 16384$ | $L = 2$ | 200 | 20000 | $(1, 40, 0)$ | 0.4 |
| $q = 32768$ | $L = 2$ | 200 | 40000 | $(1, 40, 0)$ | 0.4 |
| | $L = 3$ | 200 | 40000 | $(1, 40, 40)$ | 0.4 |
| $q = 8192$ | $L = 2$ | 400 | 4000 | $(1, 20, 0)$ | 0.4 |

Tables 11 and 12 give the optimal hyperparameters found for linear probing on IN100%, linear probing on Places205, and semi-supervised learning with limited labels of IN100% when evaluating pretrained ResNet-50 backbones with SFRIK and VICReg. The hyperparameters that are tuned for evaluation are: the initial learning rate $lr\_head$ of the linear layer in linear probing; and the initial learning rate $lr\_backbone$ and $lr\_head$ for respectively the backbone and the linear layer in semi-supervised learning. The reported hyperparameters in the two tables yield the evaluation results reported in Section 4.3.

## C.7 Computational resources

Pretrainings of ResNet-18 on IN20% with a batch size of 2048 (respectively 4096) are performed with 4 (respectively 8) NVIDIA Tesla V100 GPUs with 32GB of memory each. Pretrainings of ResNet-50 on IN100% are performed with 8 NVIDIA Tesla V100 GPUs with 32GB of memory each. The total amount of compute used for this work is around 50000 GPU hours.

Table 11: Hyperparameter tuning for linear probing on IN100% and Places205 for SFRIK and VICReg pretrained on IN100% with a ResNet-50.

| | | $lr\_head$ | |
|---|---|---|---|
| Method | Epochs | IN100% | Places205 |
| VICReg$^\dagger$ ($q = 8192$) | 200 | 0.02 | 0.01 |
| SFRIK ($L = 2, q = 8192$) | 200 | 1.0 | 0.01 |
| SFRIK ($L = 3, q = 8192$) | 200 | 2.0 | 0.01 |
| SFRIK ($L = 2, q = 16384$) | 200 | 0.3 | 0.01 |
| SFRIK ($L = 2, q = 32768$) | 200 | 1.0 | 0.01 |
| SFRIK ($L = 3, q = 32768$) | 200 | 0.4 | 0.01 |
| SFRIK ($L = 2, q = 8192$) | 400 | 2.0 | 0.01 |

Table 12: Hyperparameter tuning for semi-supervised learning for SFRIK and VICReg pretrained on IN100% with a ResNet-50.

| | | Semi-supervised, 1% labels | | Semi-supervised, 10% labels | |
|---|---|---|---|---|---|
| Method | Epochs | $lr\_backbone$ | $lr\_head$ | $lr\_backbone$ | $lr\_head$ |
| VICReg$^\dagger$ ($q = 8192$) | 200 | 0.02 | 0.2 | 0.2 | 0.04 |
| SFRIK ($L = 2, q = 8192$) | 200 | 0.004 | 1.6 | 0.02 | 0.4 |
| SFRIK ($L = 3, q = 8192$) | 200 | 0.002 | 1.0 | 0.01 | 0.2 |
| SFRIK ($L = 2, q = 16384$) | 200 | 0.004 | 1.4 | 0.04 | 0.2 |
| SFRIK ($L = 2, q = 32768$) | 200 | 0.004 | 1.0 | 0.04 | 0.1 |
| SFRIK ($L = 3, q = 32768$) | 200 | 0.004 | 1.0 | 0.02 | 0.1 |
| SFRIK ($L = 2, q = 8192$) | 400 | 0.004 | 1.4 | 0.02 | 0.2 |

## C.8 PUBLIC RESOURCES

We acknowledge the use of the following public resources, during the course of the experimental work of this paper:

- VICReg official code (Bardes et al., 2022) . . . . . . . . . . . . . . . . . . . . . . . . . . . . . . MIT License
- DINO official code (Caron et al., 2021) . . . . . . . . . . . . . . . . . . . . . . . . . . . Apache License 2.0
- OBoW official code (Gidaris et al., 2021) . . . . . . . . . . . . . . . . . . . . . . . . Apache License 2.0
- SwAV official code (Caron et al., 2020) . . . . . . . . . . . . . . . . . . . . . . . . . . . . . CC BY-NC 4.0
- SimCLR official code (Chen et al., 2020a) . . . . . . . . . . . . . . . . . . . . . . . . Apache License 2.0
- VISSL code (Goyal et al., 2021) . . . . . . . . . . . . . . . . . . . . . . . . . . . . . . . . . . . . . MIT License
- ImageNet dataset (Deng et al., 2009) . . . . . . . . . . . . . . . . . . . . . . . . . . . . . . . . . . . . . . . . . .
- Places 205 dataset (Zhou et al., 2014) . . . . . . . . . . . . . . . . . . . . . . . . . . . . Attribution CC BY
- VOC2007 dataset (Everingham et al., 2010) . . . . . . . . . . . . . . . . . . . . . . . . . . . . . . . . . . . . .

# D    ADDITIONAL EXPERIMENTAL RESULTS

We provide in this appendix other experimental results to complement the main paper.

## D.1    HYPERPARAMETER TUNING WITHOUT A SEPARATE VALIDATION SET

A common practice in the self-supervised learning literature, e.g., (Bardes et al., 2022; Chen et al., 2020a), is to select the hyperparameters by measuring the performance on the validation set of ImageNet (red dataset in Figure 2) instead of a separate validation dataset (purple dataset in Figure 2). In this paragraph, we verify whether this less rigorous practice changes the conclusion of the experiments in Section 4.2. In Table 13, we report the evaluation of the different backbones pretrained on IN20% after tuning each method *directly on the validation set* of ImageNet, which is the same dataset used for evaluation in linear probing. By comparison with Table 2, which follows the more rigorous hyperparameter tuning protocol described in Appendix C.4, we observe that although the absolute figures of merit slightly vary if we use the less rigorous protocol instead of the more rigorous one, the conclusion of the experiments in Section 4.2 does not change. This gives an empirical justification to this common practice.

Table 13: **Linear probing on IN20% (top-1 accuracy)** at different embedding dimensions $q$. All methods were pretrained on IN20% with a ResNet-18 for 100 epochs. Hyperparameters specific to each method and the learning rate are tuned on the *same* dataset as the one used for evaluation in linear probing, which is less rigorous than tuning the hyperparameters on a separate validation set as described in Appendix C.4. Symbol † indicates models that we retrained ourselves.

|             | SimCLR† | AUH† | VICReg† | SFRIK $L = 1$ | SFRIK $L = 2$ | SFRIK $L = 3$ |
|-------------|---------|------|---------|---------------|---------------|---------------|
| $q = 1024$  | 45.2    | 45.2 | 40.8    | -             | 44.2          | -             |
| $q = 2048$  | 45.8    | 45.6 | 44.1    | -             | 45.5          | -             |
| $q = 4096$  | 46.3    | 46.8 | 44.9    | -             | 47.0          | -             |
| $q = 8192$  | 46.2    | 46.8 | 46.0    | 27.5          | 47.0          | **47.6**      |

## D.2    ADDITIONAL EVALUATION OF SFRIK PRETRAINED ON IN20%

In complement to the results on linear probing on IN20% presented in Table 2, we evaluate SFRIK by linear probing on Places205 (Zhou et al., 2014) and linear SVM on VOC2007 (Everingham et al., 2010), following the protocol described in Appendix C.3. The learning rate for linear probing on Places205 is fixed at 0.01. Table 14 compares SFRIK, AUH and VICReg pretrained with ResNet-18 on IN20% under the setting of Section 4.2.

Table 14: **Linear classification on Places205 and VOC2007 (accuracy and mean average precision).** All methods were pretrained on IN20% with a ResNet-18 for 100 epochs. We tuned all hyperparameters specific to each pretraining method and the learning rate. The symbol † indicates models that we retrained ourselves.

| Method | Places205 Top-1 | Places205 Top-5 | VOC07 mAP |
|--------|-----------------|-----------------|-----------|
| VICReg† ($q = 8192$)        | 41.6 | 71.7 | 73.3 |
| AUH† ($q = 8192$)           | 42.3 | 72.8 | 73.6 |
| SFRIK ($L = 3, q = 8192$)   | **42.7** | **72.9** | **74.1** |

We conclude that, under the rigorous protocol of Section 4.2 and Appendix C.4, SFRIK performs better than AUH on various downstream tasks, while having the same computational saving offered by the kernel trick. Compared to VICReg, SFRIK performs better on these tasks with a reduced complexity.

## D.3 Results of other pretraining methods on IN100% with ResNet-50

In complement to Table 4, Table 15 reports evaluation results for linear probing on IN100% and semi-supervised learning with few labels of IN100% of different ResNet-50 pretrained on IN100% with other state-of-the-art methods than the ones presented in Table 4. As mentioned in Section 4.3, we observe that, similarly to SFRIK, both BYOL and SwAV with multi-crop have a performance drop compared to VICReg on semi-supervised learning with 1% of labels, even though they perform better on linear probing on IN100%.

Table 15: **Linear probing on IN100%, semi-supervised learning with few labels of IN100% (top-1 and top-5 accuracy).** All methods have been pretrained on IN100% with a ResNet-50 during the reported number of epochs.

| Method | Epochs | Linear probing IN100% | | Semi-supervised 1% labels | | 10% labels | |
|---|---|---|---|---|---|---|---|
| | | Top-1 | Top-5 | Top-1 | Top-5 | Top-1 | Top-5 |
| SimCLR (Chen et al., 2020a) | 1000 | 68.3 | 89.0 | 48.3 | 75.5 | 65.6 | 87.8 |
| OBoW (Gidaris et al., 2021) | 200 | 73.8 | - | - | 82.9 | - | 90.7 |
| BYOL (Grill et al., 2020) | 1000 | 74.3 | 91.6 | 53.2 | 78.4 | 68.8 | 89.0 |
| SwAV (with multi-crop) (Caron et al., 2020) | 800 | 75.3 | - | 53.9 | 78.5 | 70.2 | 89.9 |
| Barlow Twins (Zbontar et al., 2021) | 1000 | 73.2 | 91.0 | 55.0 | 79.2 | 69.7 | 89.3 |
| VICReg (Bardes et al., 2022) | 1000 | 73.2 | 91.1 | 54.8 | 79.4 | 69.5 | 89.5 |

