# OpenReview forum: "Self-supervised learning with rotation-invariant kernels"
_ICLR.cc/2023/Conference — ICLR 2023 notable top 25%_

### Official Review · Reviewer_7yz3 · 2022-10-23

**Confidence:** 3
**Correctness:** 4
**Technical Novelty And Significance:** 2
**Empirical Novelty And Significance:** 2
**Recommendation:** 8

**Clarity, Quality, Novelty And Reproducibility:**

This paper is written well and is well motivated. It is relatively easy to follow the authors’ reasoning, though background knowledge of SSL methods and in particular VICReg is necessary for a smooth reading.

The provided mathematics should allow for a high-level of reproducibility, though no code release is promised in the submitted manuscript.

The work is quite original and seems to be mathematically sound.

**Strength And Weaknesses:**

The proposed method is quite elegant, and can be understood reasonably well in an intuitive manner. The related works are positioned and explained well.

The method-wise contribution of this paper can be summarized as the use of a truncated kernel for the uniformity loss (called SFRIK). Though the mathematical formalization of the loss formulation and the rigorous discussion of related work is very much appreciated, one must wonder about the simple question: are the proposed changes demonstrated in a convincing manner. While the improved RAM and training time (shown in Tab. 5) are great, the main results shown in Tab. 4 does not paint a convincing picture as the proposed method mostly reaches performance parity in linear probing and does not perform well in the semi-supervised 1% case.

The authors perform their main experiments on a benchmark they dub “IN20%”. This is not a standard benchmark, so the reviewer would like to ask about the rationale for selecting this setting. Furthermore, despite the heavy comparisons to VICReg, many experiments are missing that were conducted in the VICReg paper (such as VOC07, iNat18, COCO det+seg, MS-COCO retrieval). While not all experiments should be required, one wonders if there are sufficient tasks being considered to argue for the general applicability of the proposed method.

**Summary Of The Paper:**

This paper tackles the problem of self-supervised learning without negative pairs (such as that used in InfoNCE) and defines a so-called “loss family” which expresses existing methods such as VICReg as a specific case of the family. The paper also proposes the use of a truncated kernel which results in a linear scaling of the loss complexity in relation to the SSL embedding dimension (as opposed to quadratic for VICReg). The proposed method shows respectable performances in comparison to the state-of-the-art, while requiring less memory and training time compared to VICReg in particular.

**Summary Of The Review:**

The paper was a pleasure to read through, and the formalization of uniformity in instance-wise SSL was done well. The experimental results are reasonably convincing, but fall short when compared to similar works in literature. I would like to read the comments from the other reviewers, as well as the rebuttal from the authors before making a more certain recommendation.


**Edit after rebuttal period**

I have carefully gone through the other reviews as well as the responses provided by the authors. I believe that the authors' responses are rigorous, earnest and sufficient. I agree in spirit with the authors that pre-training on IN100% with ResNet-50 is certainly excessively expensive when considering a fair and sufficiently hyperparameter-tuned experimental setting. Thus, I raise my rating to 8.

---

> ### Author Response · Authors · 2022-11-14
> **Rebuttal to reviewer 7yz3**
>
> We thank the reviewer for taking the necessary time and effort to review our manuscript. We address here the reviewer's comments as follows:
>
> > Reason for IN20%?
>
> As SimCLR, AUH and VICReg were originally introduced with different settings, e.g. different image augmentations or projection head architecture, we decided to reimplement these methods under the same experimental setting of Section 4.1, in order to perform a fair comparison with SFRIK. Consequently, such a protocol requires rigorous hyperparameter tuning of each method. But due to the computationally intensive nature of pre-training on IN100% with ResNet-50, we decided to pretrain methods on IN20%, originally introduced in (Gidaris et al., 2021), with ResNet-18 in order to perform complete hyperparameter tuning of each method with a reasonable amount of computing resources. Furthermore, using only 20 % of the training set of ImageNet for pretraining allows for designing a *separate* validation set for a valid hyperparameter tuning, in which the test set for final evaluation is never seen during tuning.
>
> In addition, compared to other reduced versions of ImageNet-1k with fewer classes like ImageNet-100, we believe that IN20% is more challenging for pretraining, since it keeps all the 1000 classes of ImageNet-1k (but with only 260 images per class), which is why we think that IN20% is more relevant for benchmarking than ImageNet-100.
>
> > Are the proposed changes demonstrated in a convincing manner?
>
> On the IN20% benchmark, all methods are compared in the fairest way, with a hyperparameter tuning on a separate validation set. Under this setting, Table 2 shows that SFRIK outperforms SimCLR, AUH and VICReg on IN20% linear probing. In our answer to reviewer uDSt, we further illustrate the superiority of SFRIK in this fair setting on two additional downstream tasks (linear probing on Places205 and linear SVM on VOC2007).
>
> On the common evaluation protocol (IN100%, ResNet-50, 200 epochs), we claim that SFRIK has a higher performance on several downstream tasks compared to VICReg, while having a smaller computational complexity during pretraining. To support this claim, we provide in this rebuttal the evaluation results of *a new pretrained ResNet-50 with SFRIK at truncation order $L=3$ and embedding dimension $q=8192$* using the same protocol of Section 4.3.
>
> Firstly, for pretraining on a machine with 8 GPUs having 32GB of memory each, SFRIK (L=3, q=8192) is 8% faster than VICReg (q=8192) and needs 3% less memory per GPU. In addition to this computational gain, the following table shows that SFRIK has better evaluation results than VICReg on several downstream tasks (except for semi-supervised learning with 1% of labels), including linear probing on IN100%, linear probing on Places205, and semi-supervised with 10% of labels.
>
> ||Linear probing IN100%|Linear probing Places205|Semi-supervised 1%|Semi-supervised 10%|
> |-|-|-|-|-|
> |VICReg (q=8192)|70.0 / 89.3|54.1 / 83.4|**49.4 / 75.1**|65.9  / 87.2|
> |SFRIK (L=3 q=8192)|**70.2 / 89.6**|**54.5 / 83.9**|46.9 / 73.6|**66.0 / 87.7**|
>
> As explained in Section 4.3, the performance drop on semi-supervised learning with 1% of labels is not specific to SFRIK because we observe the same drop with other methods like BYOL or SwAV (see Appendix C.9). Therefore, we hypothesize that there exists an ingredient in VICReg that makes it more robust on this specific downstream task (e.g., the fact that VICReg does not normalize embeddings), and the study of such a conjecture is left as a future work.
>
> > Additional downstream task?
>
> We provide in this rebuttal some complements to Table 4 by showing evaluation results on an *extra downstream task*, in addition to linear probing on IN100%, Places205, and semi-supervised learning with few labels. Following the protocol of (Bardes et al., 2022), we evaluate SFRIK and VICReg pretrained with ResNet-50 on IN100% during 200 epochs by linear SVM on VOC2007.
>
> Table: Mean average precision (mAP). Features are extracted after the last average pooling of ResNet-50.
> ||Linear SVM VOC2007|
> |-|-|
> |VICReg (q=8192)|84.9|
> |SFRIK (L=2, q=8192)|85.1|
> |SFRIK (L=3, q=8192)|84.6|
> |SFRIK (L=2, q=16384)|**85.2**|
> |SFRIK (L=2, q=32768)|85.0|
> |SFRIK (L=3, q=32768)|85.1|
>
> While having a reduced pretraining complexity, SFRIK (L=2, q=8192) surpasses VICReg (q=8192) on VOC2007 linear SVM, which argues in favor of the general applicability of SFRIK. The highest score is achieved by SFRIK at q=16384, which illustrates the computational advantage of SFRIK over VICReg, because, as opposed to VICReg, SFRIK is able to scale at large embedding dimension to improve performance (at q=16384, SFRIK is 19% faster than VICReg during pretraining and needs 8% less memory per GPU for a machine with 8 GPUs having 32GB memory each).
>
> > Code release?
>
> Code is not provided during submission due to difficulties of guaranteeing anonymity, but we commit to release the code at publication in the interest of reproducible research.

---

### Official Review · Reviewer_w91x · 2022-10-25

**Confidence:** 4
**Correctness:** 4
**Technical Novelty And Significance:** 4
**Empirical Novelty And Significance:** 3
**Recommendation:** 8

**Clarity, Quality, Novelty And Reproducibility:**

The paper is very clearly motivated and presented.
The method is theoretically supported and the thoroughly empirically benchmarked.
The method is novel and solidifies connections between methods putting them on solid theoretical footing.
The authors will provide code to reimplement their results, which meets the gold standard of Reproducibility in the community.

**Strength And Weaknesses:**

This paper has no obvious major weaknesses and several strengths:
- The construction of the method is based on solid theoretical foundation connecting to kernel methods, integral probability metrics (MMD) and spherical harmonics.
- The paper established interesting and wide-reaching connections to other methods, unifying uniformity-based self-supervised learning and information-maximization methods.
- SFRIK enjoys different complexity scalings compared to for instance VICReg thanks to its use of the kernel trick, making it clearly advantageous in specific regimes like large expander dimension and small batch size.
- The paper delves into rigorous empirical comparisons with the major competing methods (SimCLR, SwAV, SimSiam, VICReg), fixing architecture and probing different embedding dimensions on multiple downstream tasks of interest (linear probing, kNN classification, semi-supervised learning), and even comparisons with teacher-student methods like BYOL.
- The paper includes a rigorous study on the effect of the main hyperparameters, such as embedding dimension, kernel type and kernel truncation order.

**Summary Of The Paper:**

The paper deals with self-supervised learning and proposes a method named SFRIK that complements a Siamese architecture with a regularization term that consists in the MMD discrepancy between the embedding distribution and the uniform distribution on the hypersphere. In essence, the idea of this term is to prevent representation collapse by making sure that the learned embeddings are uniformly spread out across the hypersphere.
The methods interestingly unifies and subsumes several previously proposed uniformity-based self-supervised learning procedures, including recent ones such as VICReg, and connects to information-maximization methods.
The paper exhaustively examines the effect of the various hyperparameters (such as expander dimensionality, kernel, and kernel truncation order) of SFRIK and compares its performance in terms of quality of the learned representations to those learned by the main competing algorithms like SimCLR, SwAV and VICReg on multiple downstream on ImageNet1k and Places205. The conclusions are that SFRIK is competitive with these algorithms both in terms of quality of the learned embeddings for downstream classification tasks, as well as in training speed and memory efficiency.

**Summary Of The Review:**

Very interesting and relevant work proposing a new class of self-supervised learning regularizations, based on solid theoretical foundation with possibly wide-reaching implications, even beyond the convincing empirical results already presented in the paper.

---

> ### Author Response · Authors · 2022-11-14
> **Rebuttal to w91x**
>
> We thank the reviewer for taking the necessary time and effort to review our manuscript, and his positive feedback. The reviewer did not raise any obvious major weakness in the paper therefore we do not further comment on the review in this rebuttal.

---

### Official Review · Reviewer_uDSt · 2022-10-25

**Confidence:** 3
**Correctness:** 3
**Technical Novelty And Significance:** 2
**Empirical Novelty And Significance:** 2
**Recommendation:** 6

**Clarity, Quality, Novelty And Reproducibility:**

The clarification of the novelty part ma need some revision.



**Strength And Weaknesses:**

Pros:
The author proposed SFRIK for avoiding representation collapse in invariant representation learning.
The author did thorough study on the truncated kernel and drew connection between the proposed method and information maximization.
The author conducted experiments for both representation quality and runtime complexity measurement.



Cons:
1. The author proposed a family of losses which are powered by different choice of kernels. However, the choice of RBF kernel and generalized distance kernel have already been studied in existing works. as the author mentioned in table 1. Hence the novelty of the proposed method shrunk to the adaptation of truncated kernel.
2. The author mentioned that the proposed SFRIK can help saving computation and memory complexity comparing with VICReg due to the kernel trick. However, applying RBF kernel should provide the same saving. As the author mentioned adopting RBF kernel is nearly equivalent to the AUH method. The saving on computational complexity shouldn’t be considered as the novelty of SFRIK.
3. In the experiment section. Table 2 shows AUH method performs quite competitive to the proposed method. However, in Table 3, the author show the performance of applying RBF kernel is much worse than the proposed truncated kernel. The author may add some explanation of the discrepancy if AUH works equivalent as applying RBF kernel.
4. The motivation of applying the truncated kernel was not clear enough. Though this kernel function was not investigated in self-supervision learning. The author may explain more on why the adoption is necessary.

**Summary Of The Paper:**

The authors proposed a generic regularization loss on self-supervision learning of image representations. The regularization is proposed to enforce the uniformity of transformed representations hypersphere. The author show that the proposed loss family accommodates several previous self-supervision learning methods. Moreover, the author emphasized that the proposed regularization with truncated kernel can significantly save memory and runtime. Experiment results show the advantages of applying SFRIK in self supervision learning.

**Summary Of The Review:**

Overall, the paper needs some clarification of the motivation and novelty.

---

> ### Author Response · Authors · 2022-11-14
> **Rebuttal to reviewer uDSt**
>
> We thank the reviewer for taking the necessary time and effort to review our manuscript. We address here the reviewer's comments as follows:
>
> > The novelty of the proposed method shrinks to the adaptation of a truncated kernel.
>
> The paper's novelty includes the adaptation of truncated kernels for self-supervision, but it is not reduced to it. As mentioned by reviewers 3r4M, w91x and 7yz3, one major novelty of the paper is the unification of several previous self-supervision methods under the proposed generic regularization loss (7). This general framework allows us to understand that former regularizers are in fact variants of (7) for different kernel choices. Therefore, the paper suggests that the design of a self-supervision method can be reduced to the question of choosing a kernel that yields good representations for downstream tasks, which in itself is a novel research direction. In this context, Section 4 provides a rigorous empirical comparison between the truncated kernel and previous kernels used in the literature that can help understanding how to choose a good kernel for self-supervision. In other words, one important contribution of the paper is to study self-supervision under a new unified kernel point of view that can provide novel insights about the behavior of self-supervision algorithms.
> > The saving on computational complexity shouldn’t be considered as the novelty of SFRIK.
>
> The reviewer is right to say that the RBF kernel provides the same saving as the truncated kernel. Therefore, the main novelty of SFRIK compared to AUH is its higher performance on downstream tasks with the same computational saving as AUH (we recall that the complexity of VICReg is quadratic in the embedding dimension, but linear for SFRIK and AUH).
>
> According to Table 2, SFRIK (L=3) has a higher top-1 accuracy for linear probing on IN20% compared to AUH. In order to further illustrate the improved performance of SFRIK over AUH, we provide below *new evaluation results for linear SVM on VOC2007*, *and linear probing on Places205*, following the protocol of (Bardes et al., 2022) and Appendix C.3.
>
> Table: Top-1 accuracy for IN20% and Places205, mean average precision for VOC2007. Features for linear SVM are extracted after the last average pooling of ResNet-18.
> ||Linear probing IN20%|Linear probing Places205|Linear SVM VOC2007|
> |-|-|-|-|
> |VICReg (q=8192)|46.0|41.6|73.3|
> |AUH (q=8192)|46.8|42.3|73.6|
> |SFRIK (L=3, q=8192)|**47.5**|**42.7**|**74.1**|
>
> We conclude that, under the rigorous protocol of Section 4.2 and Appendix C.4, SFRIK performs better than AUH on various downstream tasks, while having the same computational saving offered by the kernel trick. Compared to VICReg, SFRIK performs better on these tasks with a *reduced* complexity.
>
> > Explanation of the discrepancy between AUH and RBF kernel?
>
> The uniformity loss (7) with the RBF kernel is $\ell_u(\mathbf{Z}_I)=\frac{1}{|I|^2}\sum _{i,i'\in I}e^{-t ||\mathbf{z}_i-\mathbf{z} _{i'}||^2}$, while the uniformity loss of AUH is $\log(\ell_u(\mathbf{Z}_I))$. Therefore the observed performance discrepancy between these two terms is only due to the presence of the logarithm in AUH. But as we do not fully understand the importance of such a design for self-supervision, we clarify in the revision that the study of the logarithm before the uniformity term (7) is left as a future work.
>
> > Motivation behind the choice of a truncated kernel?
>
> The motivation is to understand what ingredients are important for a good kernel choice in self-supervision. Our general framework for regularization considers rotation-invariant kernels because they are used in statistical tests for uniformity (Garcia-Portugues & Verdebout, 2018). Since there are many possible choices of such kernels, we analyze the choice of a good kernel with the characterization of Theorem 1, which expands any rotation-invariant kernel into a non-truncated Legendre expansion with nonnegative kernel weights.
>
> Under this view, Section 4 studies empirically the impact of the first terms of the Legendre expansion, using the so-called truncated kernel. By comparing this truncated kernel with previous non-truncated kernels, our experiments show that truncated kernels at order $L=2, 3$ already provide competitive results compared to the state of the art. *This illustrates the importance of the first terms in the Legendre expansion, because they are sufficient to explain the performance of self-supervised learning algorithms.*
>
> This observation is in fact consistent with the empirical results of information-maximization methods based on the covariance matrix (Bardes et al., 2022; Ermolov et al., 2021), since the kernel mean embedding associated to a truncated kernel at order $L=2$ encodes the first two moments of a distribution (cf. Section 3.3). An original extension proposed by the paper is to also capture the third moment by choosing $L=3$, and Section 4 indeed demonstrates the benefit of this choice.

---

> > ### Comment · Reviewer_uDSt · 2022-12-03
> > **Thanks for the response**
> >
> > I appreciate the detailed response from the authors, especially the additional experiment results to raise the convincingness of this work. I'm glad to see the authors are willing to continue study the possible reason of performance discrepancy between AUH and RBF kernel. I thus raise my recommendation score to 6.

---

### Official Review · Reviewer_3r4M · 2022-10-25

**Confidence:** 3
**Correctness:** 3
**Technical Novelty And Significance:** 4
**Empirical Novelty And Significance:** 3
**Recommendation:** 6

**Clarity, Quality, Novelty And Reproducibility:**

The paper is well written. The idea reflected in the paper seems to be novel to me.

**Strength And Weaknesses:**

Strength:

- This paper introduces a generic regularization loss based on kernel trick and might be of interest of the community to better understand self-supervised method from a kernel perspective.

- The introduced loss family uniforms several mainstream self-supervised methods. This includes uniformity-based and information-maximization methods as a special case of the proposed framework under different kernel choices.

- The method is able to reduce computation time and memory efficiently with the help of kernel trick while it remains competitive on downstream tasks


Weakness:
- W1:  Why L is only constrained to be 2 and 3? What happens when L becomes bigger? Is there any relevant ablation studies on this?

- W2: Is it possible to explain the relationship between different kernel choices and the eventual feature quality? How does different kernel choices affect downstream task performance? I asked this because it seems they all have the same goal of pull the embedding distribution to be close to uniform distribution. How can we know that which choice of kernel is the best among all other kernels? Also, it seems you don't achieve SOTA performance on all settings.

- W3: It might be helpful to give a brief introduction about kernel mean embeddings as preliminaries to improve readability for those who are not familiar with this topic.


**Summary Of The Paper:**

Summary:
This paper proposes a regularization loss by resorting to rotation-invariant kernels. The method promotes embedding distribution to be close to uniform distribution in order to avoid collapse in self-supervised learning.



**Summary Of The Review:**

Given the novelty, technical and theoretical contribution of the paper, I recommend acceptance of the paper.

---

> ### Author Response · Authors · 2022-11-14
> **Rebuttal to reviewer 3r4M**
>
> We thank the reviewer for taking the necessary time and effort to review our manuscript. We address here the reviewer's comments as follows:
>
> > W1: Why L is only constrained to be 2 and 3? What happens when L becomes bigger? Is there any relevant ablation studies on this?
>
> According to Table 2, the gain in top-1 accuracy by linear probing on IN20%  between SFRIK at $L = 1$ and $L = 2$ is important (+19.3%), but is smaller between $L = 2$ and $L = 3$ (+1.5%). This suggests that $L > 3$ is likely to marginally improve performance. Exploring $L > 3$ is possible, but for fair comparison it requires additional hyperparameters tuning (there is one more kernel weight to tune for each additional order). Therefore we limit our study at truncation order $L \leq 3$, which is sufficient because SFRIK at $L=2$ and $L=3$ already provides competitive results compared to the state of the art, as illustrated in Tables 2 and 4.
>
> > W2: Is it possible to explain the relationship between different kernel choices and the eventual feature quality? How does different kernel choices affect downstream task performance? I asked this because it seems they all have the same goal of pull the embedding distribution to be close to uniform distribution. How can we know that which choice of kernel is the best among all other kernels?
>
> The reviewer is right to say that the goal of the generic regularization loss (7) is to pull the embedding distribution to be close to the uniform distribution. However, to achieve this goal, the choice of the kernel is crucial to quantify in a relevant way the divergence between the embedding distribution and the uniform distribution on the hypersphere.
>
> To illustrate this claim, we argue that there exist some bad kernel choices, like the linear kernel $k(\mathbf{u}, \mathbf{v}) = \mathbf{u}^{\top} \mathbf{v}$, for which the kernel mean embedding mapping loses too much information about the distribution, and makes the regularization term (7) not strong enough to avoid a certain form of collapse. Indeed, the kernel mean embedding associated to the linear kernel is:
>
> $ \mathbf{v} \mapsto \mathbb{E} _{\mathbf{u} \sim \mathbb{Q}} [ \mathbf{u}^{\top} \mathbf{v} ]  = \mathbb{E} _{\mathbf{u} \sim \mathbb{Q}} [ \mathbf{u}]^{\top} \mathbf{v} $
>
> for any distribution $\mathbb{Q}$ on the hypersphere. This kernel mean embedding is the linear form that can be identified (by Riesz representation theorem) with $\mathbb{E}_{\mathbf{u} \sim \mathbb{Q}} [ \mathbf{u}]$, i.e., the expectation of $\mathbb{Q}$. Therefore, the regularization term (7) with a linear kernel only enforces the embedding distribution to have a zero expectation. This is not sufficient to avoid a dimensional collapse (Jing et al., 2022), since a batch of embeddings laying in a low-dimensional subspace with a zero mean would satisfy the constraint of the regularization loss, but would be undesirable for learning good representations. Numerically, we indeed observe in Table 2 that choosing the truncation order $L=1$ in SFRIK, which is exactly the same as choosing a linear kernel in (7), yields poor results on linear probing.
>
> As suggested by the reviewer's question, the theoretical characterization of kernels that yield high-quality learned representations during pretraining is in itself an interesting future research direction. Finding theoretical arguments explaining why a kernel is better than another would indeed provide guarantees to self-supervised training via our general kernel framework.
>
> As a first step toward this research direction, our paper provides an empirical study of the impact of different kernel choices on the quality of the learned representations. We indeed show experimentally (cf. Section 4.2) in a rigorous setting that a truncated kernel at low order $L \leq 3$ is a better choice than the RBF and the generalized distance kernel, whose kernel weights $b_\ell$ in the Legendre expansion decreases at least in a polynomial rate with the order $\ell$. These empirical observations can help future theoretical investigation of a good kernel choice.
>
> > W3: It might be helpful to give a brief introduction about kernel mean embeddings as preliminaries to improve readability for those who are not familiar with this topic.
>
> We agree with the reviewer that giving a brief introduction about kernel mean embeddings can improve readability of the manuscript, and we thank the reviewer for this suggestion. As the number of pages for the main text is limited, we introduce reminders about kernel mean embeddings in the supplementary materials of the revised manuscript (Appendix A).

---

> > ### Comment · Reviewer_3r4M · 2022-11-25
> > **Thanks for your reply**
> >
> > I appreciate your detailed replies and additional explanations I therefore maintain my scores.

---

### Author Response · Authors · 2022-11-14
**Revised manuscript**

Following the reviewers' feedback on our submission, we propose a revision of the manuscript with the changes indicated in color to accompany our rebuttal.

---

### Decision · Program_Chairs · 2023-01-20

**Decision:**

Accept: notable-top-25%

**Justification For Why Not Higher Score:**

Though the results are interesting and novel, the core proposals (regularization loss based on kernel mean embeddings with rotation-invariant kernels on the hypersphere, truncated kernel) build on earlier works. These are thus not at the level of groundbreaking results.



**Justification For Why Not Lower Score:**

This paper was very well received by the reviewers. The paper has nicely put together existing results and proposes an interesting approach.

**Metareview: Summary, Strengths And Weaknesses:**

The paper introduces a regularization loss based on kernel mean embeddings with rotation-invariant kernels on the hypersphere for self-supervised learning of image representations.  The regularization enforces the uniformity of transformed representations on hypersphere to prevent feature collapse.  As part of the strengths, the proposed regularization with a truncated kernel can also save memory and runtime.

Reviews of this paper are mostly positive. The use of a positive definite kernel for self-supervised learning provides a new perspective (3r4M). The proposal unifies existing approaches (3r4M, uDSt, w91x). The paper is well written (3r4M), and offers a thorough empirical study (7yz3). Concerns raised were on poor performance of the RBF kernel, and lesser performance gains in the semi-supervised setting. These were resolved with rebuttals which provided more experimental results, convincing the reviewers.

Recommendation: accept.

*Suggestion from the AC*:
The use of positive definite kernels in self-supervised learning makes this work closely related to “Self-supervised learning with kernel dependence maximization”, NeurIPS 2021. I strongly suggest that the authors make a qualitative comparison to it. The aspects of encouraging uniformity on a hypersphere, unifying some existing methods, and the use of MMD (as opposed to HSIC) provide some distinction. However, considering that HSIC can be interpreted as MMD, this work must be better contrasted.


**Note From Pc:**

if the above contains the word "oral" or "spotlight" please see: "oral" presentation means -> notable-top-5% and "spotlight" means -> notable-top-25%. As stated in our emails, we are disassociating presentation type from AC recommendations

**Summary Of Ac-Reviewer Meeting:**

N/A